# Scaleable Quantum Control via Physics Constrained Reinforcement Learning

## Abstract

Quantum optimal control is concerned with the realisation of desired dynamics in quantum systems, serving as a linchpin for advancing quantum technologies and fundamental research. Analytic approaches and standard optimisation algorithms do not yield satisfactory solutions for large quantum systems, and especially not for real world quantum systems which are open and noisy. We devise a physics-informed Reinforcement Learning (RL) algorithm that restricts the space of possible solutions. We incorporate priors about the desired time scales of the quantum state dynamics – as well as realistic control signal limitations – as constraints to the RL algorithm. These physics-informed constraints additionally improve computational scalability by facilitating parallel optimisation. We evaluate our method on three broadly relevant quantum systems (multi-level $\Lambda$ system, Rydberg atom and superconducting transmon) and incorporate real-world complications, arising from dissipation and control signal perturbations. We achieve both higher fidelities – which exceed $0.999$ across all systems – and better robustness to time-dependent perturbations and experimental imperfections than previous methods. Lastly, we demonstrate that incorporating multi-step feedback can yield solutions robust even to strong perturbations.

## 1 Introduction

The optimal control of quantum systems is important for enabling the development of quantum technologies such as computing, sensing or communication, and similarly plays an important role for quantum chemistry (Brif et al., 2010) and solid state physics (Glaser et al., 2015). Quantum optimal control is concerned with the implementation of optimal external signals, applied to a quantum system, to realise desired dynamics (Glaser et al., 2015; Koch, 2016; Koch et al., 2022; Mahesh et al., 2022). Examples of such tasks include system initialisation, (quantum) state preparation, gate operation/state population transfer or state measurement. Quantum control enables performing such tasks with low error rates, which is particularly important for the realisation of fault tolerant quantum computing (Terhal, 2015). Isolated quantum systems exhibit unitary dynamics (i.e. reversible) which are comparatively easy to model for modest system sizes. Yet all real quantum systems are open, subject to some interaction with the environment and require the addition of non-unitary dynamics (i.e. irreversible) to realistically capture their evolution (Breuer & Petruccione, 2002).

Motivated by such real-world experimental setups, we address physically realistic open and dissipative quantum systems. Typically, the combination of unitary and non-unitary quantum system evolution is modelled with a controlled Gorini-Kossakowski-Sudarshan-Lindblad equation (Davies, 1974; Dirr et al., 2009) (GKSL), which is a first order linear ODE. It is also sometimes known as as the master equation, quantum Liouvillian, or Lindbladian. Solving the GKSL master equation and controlling large quantum systems is extremely computationally expensive, growing quadratically with the quantum system size, limiting the use of standard optimisation methods. Experimental imperfections and noise – arising from, e.g., signal distortion or attenuation in optical and electronic setups, or due to inherent system imperfections (Burkard, 2009) – pose additional challenges which existing approaches fail to address. In this work, we present a novel approach for controlling real-world, open quantum systems, posing quantum control as a Reinforcement Learning (RL) problem subject to physical constraints. Specifically, we learn a control policy that maximises the fidelity of the quantum control task, while removing control signals which result in overly *fast* quantum state dynamics from the space of possible solutions. A majority of quantum control tasks, including those

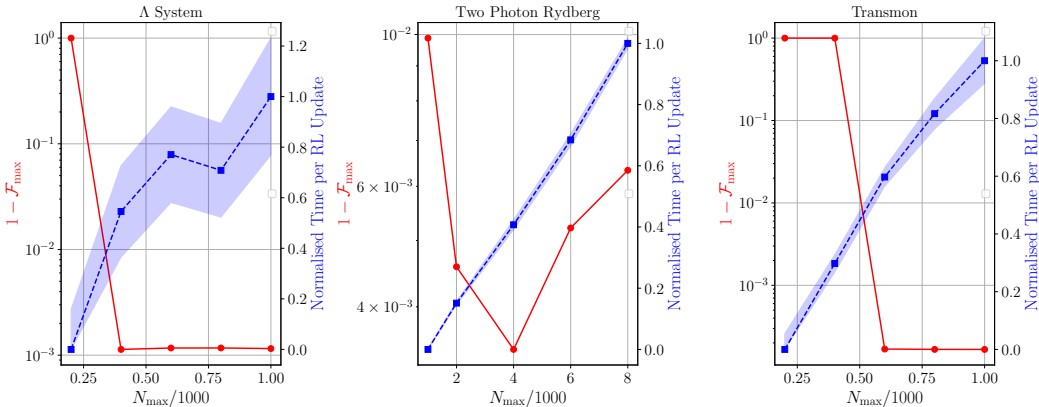

Figure 1: Infidelity for maximum fidelity across hyperparameter combinations $1 - \mathcal{F}_{\max}$ (left y-axis, solid red line) and normalised GPU time per RL update (right y-axis, dotted blue line, with one standard deviation shaded) as a function of the number of permissible quantum solver steps $N_{\max}$. We observe that limiting $N_{\max}$ – which can be understood as placing an upper bound on the rate of change of the quantum system evolution induced by the control signal – improves solution quality (lower infidelity) while also increasing computational efficiency (lower normalised time).

considered in this work, are concerned with adiabatically transferring population between quantum states (Král et al., 2007), such that the time evolution of the system is *slow* compared to the inverse energy gap of the states ($E = \hbar\omega$) which facilitate the transfer. Quantum state dynamics which are *fast* can induce leakage errors (decay outside of the desired quantum state space). Furthermore, *fast* oscillations in the state populations severely limit the robustness of control solutions to any time-dependent noise in real world experiments. In addition to the hard constraint applied to the space of possible solutions, we introduce a soft constraint that facilitates smooth pulses and fixed amplitude endpoints with finite rise-time. Both characteristics are typically required for real-world implementation of quantum control signals. Lastly, we investigate using multi-step RL to address larger levels of system noise.

Incorporating physics-based constraints into the RL problem not only enhances solution quality but also significantly improves computational scalability. In general, control signals that induce fast quantum state dynamics require complex simulations and thereby longer computation times. Excluding these signals enables fast parallel optimisation of multiple hyperparameter configurations, as control signals that would otherwise slow down the process are removed by the constraints.

We validate our approach on three quantum control problems. We begin with a generalised electronic $\Lambda$ system, common in quantum dots, atoms, colour centres, circuit quantum electro-dynamics and molecules, revisiting a well known approach (Vitanov et al., 2017) used for coherent population transfer between ground states. Our implementation successfully learns realistic control signals that outperform existing methods in terms of both fidelity $> 0.999$, and resilience to time-dependent noise. We then explore the more complex Rydberg gate (Lukin et al., 2001), crucial for realising atomic quantum computers. Here, we demonstrate robust control signals, even in the face of noise, unlike previous approaches, and achieve higher fidelities at lower pulse energy than previous works. Lastly, we consider a superconducting transmon (Egger et al., 2018b) for qubit reset, for which we discover a novel, physically-feasible reset waveform which achieves an order of magnitude higher reset fidelity than any previous work.

In conclusion, our work makes the following contributions:

1. We devise a highly scalable RL implementation that directly incorporates physical feasibility constraints to enable discovery of experimentally realistic control signals.

2. Fig. 1 demonstrates that our constraint on the maximum number of simulation steps significantly improves computational scalability while simultaneously improving solution quality.

3. Across three quantum systems, we outperform prior methods by achieving higher fidelities, lower pulse energies, and greater robustness to time-dependent noise.

| | Model Free RL (e.g. PPO) | Direct Differentiation (e.g. GRAPE) |
|---|---|---|
| **Method** | Rewards are estimated for a stochastic policy and maximised | Gradients are exactly evaluated and optimised |
| **Flexibility** | Easily handles stochasticity & multi-objective optimisation | Requires complete knowledge of time evolution of system |
| **Efficiency** | Computationally more expensive due to exploration and sampling | Faster for well-posed, deterministic problems |
| **Robustness** | Adapts to noise, parameter changes or constraints dynamically | Not robust to noise, highly sensitive to initial seed and less stable solutions |

Table 1: Comparison between Model Free RL and Direct Differentiation for optimising parameters for quantum control.

## 2 RELATED WORK

Several algorithms exist for devising optimal time-dependent control signals for quantum systems. Analytic methods like Lyapunov (Hou et al., 2012) are effective for small isolated systems but difficult to generalise to complex environments. Gradient-based methods such as GRAPE [1] (Khaneja et al., 2005) or variations of Optimal Control are efficient on smooth cost landscapes but struggle with noise and local minima for complex environments. Direct methods like CRAB [2] (Caneva et al., 2011) (sensitive to basis choice (Pagano et al., 2024)), or evolutionary algorithms (Brown et al., 2023) lack computational scalability for larger systems or multiple objectives like signal smoothness and fidelity.

Machine learning has numerous applications in quantum science (Krenn et al., 2023). We review prior work on quantum dynamic control, distinguishing between real device sampling and numerical simulations. Baum et al. (2021) devised an optimal gate set on a superconducting IBM quantum device. Reuer et al. (2023) and Porotti et al. (2022) use measurements and feedback to prepare quantum states, but generalisation is difficult. A model-based Hamiltonian learning approach was applied in Khalid et al. (2023), which does not succeed at learning time-dependent parameters and robust solutions. We show that a model-free approach with realistic noise models effectively determines interpretable, optimal signals suitable for experiments.

Several studies simulate quantum systems and apply reinforcement learning (RL) for control. RL has been applied to discrete action space control (Paparelle et al., 2020; An et al., 2021; Zhang et al., 2019), but these methods don't translate well to real-world settings with analog signals with finite response time [3] and more complex systems. We extend prior work on controlling many-body systems (Bukov et al., 2018; Metz & Bukov, 2023; Schäfer et al., 2020) to experimentally realistic systems, incorporating control signal noise into training as suggested by Schäfer et al. (2020). While Niu et al. (2019) find time-optimal gate sequences for superconducting qubits using trust region policy-gradient methods (Schulman et al., 2018), we advance this by considering experimentally realistic signals and complex noise models beyond quasi-static Gaussian errors. Our control pulses for a typical $\Lambda$ system go beyond existing work Giannelli et al. (2022a); Norambuena et al. (2023) by incorporating realistic noise models and simultaneous amplitude and frequency control to learn more optimal and realistic policies. We contrast the strengths of an RL approach compared to previous works in Tab. 1.

## 3 BACKGROUND

### 3.1 QUANTUM CONTROL

Quantum dynamics describes the time evolution of quantum systems. A system's state is represented by a *quantum state*, a vector in a complex Hilbert space $\mathcal{H}$. The most common representation is the *state vector* $|\psi\rangle \in \mathcal{H}$. A pure quantum state is described by a normalised vector (Nielsen & Chuang, 2010) $|\psi\rangle = (\psi_1, \quad \psi_2, \quad \cdots \quad \psi_n)^\top$, where $\langle\psi|\psi\rangle = 1$. A more general representation is the *density matrix* $\rho$, which for a pure state is $\rho = |\psi\rangle\langle\psi|$, (Nielsen & Chuang, 2010), and extends to classical mixtures of pure quantum states. The quantum state populations are defined as $|\psi_i|^2$ (i.e. the diagonal terms of $\rho$). Operators in quantum mechanics are unitary, making dynamics reversible. The unitary time evolution of $|\psi(t)\rangle$ is governed by the *time-dependent Schrödinger equation*:

---

[1] This stands for Gradient Ascent Pulse Engineering.

[2] This stands for Chopped Random Adiabatic Basis.

[3] A particular limitation is that of a finite rise (fall) time of an electronic or optical signal which describes the time which is required to go from zero to maximum amplitude $\geq \mathcal{O}(\text{ns})$

$$i\hbar\frac{\partial}{\partial t}\left|\psi(t)\right\rangle = \hat{H}\left|\psi(t)\right\rangle, \tag{1}$$

where $\hbar$ is the reduced Planck constant, and $\hat{H}$ is the Hamiltonian operator representing the system's total energy. Quantum control manipulates systems to achieve desired dynamics using time-dependent control fields, represented by the *control Hamiltonian*. The total Hamiltonian $\hat{H}(t)$ of a controlled system is (Giannelli et al., 2022b):

$$\hat{H}(t) = \hat{H}_0 + \sum_i a_i(t)\hat{H}_i, \tag{2}$$

where $\hat{H}_0$ is the drift Hamiltonian, $a_i(t)$ are time-dependent control actions, and $\hat{H}_i$ are control Hamiltonians. In open quantum systems, environmental interactions lead to non-unitary evolution, also sometimes described as non-coherent which makes unitary evolution coherent. The controlled Gorini-Kossakowski-Sudarshan-Lindblad equation (Davies, 1974; Dirr et al., 2009) describes this as:

$$\frac{\partial\rho(t)}{\partial t} = -\frac{i}{\hbar}[\hat{H}, \rho(t)] + \mathcal{L}(\rho(t)), \tag{3}$$

where $[\hat{H}, \rho(t)]$ denotes matrix commutation, and $\mathcal{L}(\rho)$ describes non-unitary evolution (e.g. spontaneous emission, dephasing, cavity decay, etc.). Fidelity is a common measure of similarity between quantum states. For arbitrary density matrices $\rho$ and $\sigma$, the fidelity (Jozsa, 1994) reads:

$$\mathcal{F}(\rho, \sigma) = \left(\mathrm{Tr}\sqrt{\sqrt{\rho}\sigma\sqrt{\rho}}\right)^2, \tag{4}$$

where Tr is the trace. In this paper, we evaluate the fidelity between a target state $\rho_{\mathrm{des}}$ and the final evolved state $\rho(t_f)$ to assess the effectiveness of the applied controls $a_i(t)$.

## 3.2 REINFORCEMENT LEARNING FOR QUANTUM CONTROL

Reinforcement Learning (RL) is a framework where an agent learns to make decisions by interacting with an environment to achieve a specific goal (Sutton & Barto, 1999). In quantum control, RL can be used to find the control actions $a_i(t)$ that steer a quantum system toward a target state $\rho_{\mathrm{des}}$. The key components in this RL setup are the state $(s_t)$, which is given as the density matrix $\rho_{\mathrm{fin}}(t)$ of the quantum system at the final time-step of the simulation, the control action action $(a_t)$ applied to the system, a scalar reward $(r_t)$ derived from the fidelity, indicating how close the system is to the target state, and the policy $(\pi)$ that maps states to actions. The objective is to learn a policy $\pi^*$ that maximises the expected cumulative reward over time, i.e. $\pi^* \in \max_\pi \mathbb{E}\left[\sum_{t=0}^{T} r_t\right]$.

**Bandit Setting.** In the bandit setting, the RL problem is reduced to a single time step with no state transitions. The agent selects one action $a_i$ in a continuous space $[-1, 1]$, aiming to maximise the immediate reward based on the fidelity with the target state. Specifically, the optimal action $a^*$ is given as $a^* \in \arg\max_{a_i} \mathbb{E}[r(a_i)]$, where $r(a_i)$ is the reward obtained by applying action $a_i$.

### 3.2.1 QUANTUM DYNAMICS SIMULATION

Simulating the fidelity resulting from a given control signal and initial state requires numerically solving the GKSL equation (cf. equation 3) for $\rho(t)$. This is typically done using adaptive step-size solvers that implement higher-order Runge-Kutta methods (Hairer et al., 1993), which dynamically adjust their internal time steps based on local error estimates. If the error exceeds the numerical tolerance, the solver reduces its internal time step; if the error is sufficiently small, the time step is increased to enhance computational efficiency. Therefore, control signals that lead to *slower* quantum state dynamics allow the adaptive solver to use larger time steps. Hence, they require fewer solver steps and less computation time.

## 4 METHODS

### 4.1 PHYSICS-INFORMED CONSTRAINED REINFORCEMENT LEARNING

In practice, applying reinforcement learning (RL) to find high-fidelity quantum control signals hinges on two critical aspects. First, computing the reward for the RL agent at every timestep requires simulating the quantum system (see Sec. 3.2.1). For complex quantum systems and sub-optimal actions this simulation can require an extremely large number of solver steps, which in consequence can be extremely time consuming. We remark that the compute time needed to update the RL agents can be orders of magnitude smaller than the time needed for the quantum system simulation. Second, RL optimisation algorithms are often sensitive to the choice of hyperparameters (Henderson et al., 2018), necessitating an extensive search over the hyperparameter space to find policies that achieve high or maximum fidelity.

We address the latter challenge by synchronously optimising control policies for array of up to 1024 RL agents in parallel on a single GPU device. We achieve this by implementing both the quantum solver and the RL algorithm using JAX (Bradbury et al., 2018), which features just-in-time compilation and automatic differentiation and thereby allows to compile the parallelised training and simulation loop end-to-end. However, in this parallel synchronised setup, the quantum simulation time needed per array update step is governed by the maximum quantum simulation time across all hyperparameter configurations. In other words, the slowest simulation among all learned policies determines the speed of the entire array. We mitigate this bottleneck with a physics-informed constrained RL algorithm that solves the quantum control problem subject to the condition that the required number of quantum simulation steps does not exceed a chosen threshold $N_{\max}$. Effectively, we constrain the solution space to control signals for which the quantum simulation can be executed in less then $N_{\max}$ steps. We formally define the constrained reinforcement learning problem as:

$$\pi^* \in \max_{\pi} \quad \mathbb{E}\left[\sum_{t=0}^{T} r_t\right], \text{ s.t. for } \quad a \in \pi \quad N_{\text{Sim}}(a) < N_{\text{Sim}}^{\max}, \tag{5}$$

where $\pi$ is the policy, $r_t$ is the reward at time $t$, and $N_{\text{Sim}}(a)$ is the number of solver steps required for conducting the quantum simulation for an action $a$ sampled from policy $\pi$. Implementing this constrained RL algorithm prevents bottlenecks as it ensures that all simulations within the parallelised array are completed within a fixed time frame. This approach allows us to efficiently search the hyperparameter space while maintaining computational feasibility. Although this constraint may seem restrictive, it is physically justified because we are focusing on adiabatically transferring population between quantum states (Král et al., 2007). In adiabatic processes, the system evolves slowly compared to the inverse energy gap between the states involved, which means relatively fewer solver steps are needed. The maximal effective Rabi frequency, defined as $\Omega_{\text{eff}} = \frac{\overline{\Omega}^2}{\Delta}$, gives a lower bound for the required $N_{\max}$, as a perfectly adiabatic evolution requires that according to the adiabaticity condition (Král et al., 2007) $\Omega_{\text{eff}} \cdot \delta_t \gg 1$. In practice, we increment $N_{\max}$ until a significant decrease in infidelity is observed (see Fig. 1 for infidelities at different maximum solver steps for different quantum systems).

In conclusion, the constrained RL approach not only improves computational efficiency but also promotes the selection of more physically realistic control signals. Such solutions lead to more interpretable quantum state dynamics, enhance the selection of solutions which are adiabatic in the quantum dynamics they induce and suppress spurious oscillations, thereby also promoting more experimentally realistic and robust solutions.

### 4.2 REWARD SHAPING

We parameterise the control signal(s) as a combination of time-dependent amplitudes $\Omega_i$ and time-dependent frequencies $\Delta_i$ and introduce smoothness constraints that facilitate efficient learning and further improve computational efficiency. Smoother waveforms are easier to implement experimentally, offer clearer interpretation of the optimal quantum state evolution, and significantly speed up simulation times by reducing the number of required solver steps. To facilitate smooth signal discovery, we apply a Gaussian convolution filter to our control signal with a standard deviation $t_\sigma$ (cf. App. equation 20) before simulating the quantum state dynamics which improves learning dynam-



Figure 2: Ablation over smoothness penalty coefficients $w_\Delta = w_\Omega$ and filter standard deviation $t_\sigma$ for three different environments. Choice of smoothing parameters is important for learning policies with low mean infidelity $1 - \overline{\mathcal{F}}$ (averaged over 64 parallel environments). For some systems, like the $\Lambda$ system higher filter s.d. leads to lower infidelity, whilst for other like the Transmon higher smoothing penalties lead to lower infidelities. All systems exhibit low infidelities for higher $t_\sigma$ which is important for experimental feasibility with limited bandwidth electronics.

ics by favouring slower solution dynamics (an ablation over this is found in Fig. 2). The reward function contains additional smoothing penalties and is defined as follows:

$$L = w_F \cdot \log\left(\frac{1}{1 - \mathcal{F}(\rho_{\text{fin}}, \rho_{\text{des}})}\right) - w_\Omega \cdot \text{ReLU}\left(\frac{\sum S(\Omega_i)}{\sum S_{\text{base}}} - 1\right)$$
$$- w_\Delta \cdot \text{ReLU}\left(\frac{\sum S(\Delta_i)}{\sum S_{\text{base}}} - 1\right) - w_A \cdot \frac{\sum A(\Omega_i)}{A_{\text{base}}} \tag{6}$$

The first and most important reward-function term incentives high fidelity $\mathcal{F}$ with respect to the desired final state $\rho_{\text{des}}$. This fidelity reward is proportional to $\log\left(1/(1 - \mathcal{F})\right)$. Next we define smoothness penalties, $\text{ReLu}(x)$ defines the ReLu function: $\text{ReLu}(x) = 0$ if $x < 0 \;||\; \text{ReLu}(x) = x$ if $x >= 0$. $S$ compares the smoothness of the given signal to that of a reference signal $S_{\text{base}}$ (cf. App. Sec.E.2 Fig. 13 for a definition of and ablation over different smoothing functions). We introduce a smoothness penalty weighted by $w_\Delta, w_\Omega$, to balance fidelity, interpretability, and computational efficiency. Fig. 2 shows an ablation over various smoothing penalties. Contrary to the $\Lambda$ system and Rydberg atom, the Transmon favours stronger smoothing penalties showing that our approach is adaptable to a wide variety of physical problem settings. Larger $t_\sigma$ and $w_\Delta, w_\Omega$ also reduces the maximum required solver steps and thereby further enhances computational scaleability. The ability to achieve high-fidelity solutions across all environments at larger filter standard deviations ($t_\sigma$) also demonstrates that we can find optimal signals compatible with realistic electronic control systems with limited instantaneous bandwidth.

The final reward term penalises solutions with large pulse area (cf. App. Sec. B Fig. 7 for an ablation over different area penalties for the $\Lambda$ system), we set $w_A = 0$ for the Rydberg and Transmon problem settings. We introduce additional physics-informed constraints which are problem specific and defined in App. Sec. B and App. Sec. C.

## 5 EXPERIMENTS

**Overview.** Our experiments largely focus on the bandit RL setting in a continuous action space, and are supplemented by experimentally verifying that multi-step RL is superior to the bandit setting in the presence of strong perturbations. We conducted experiments on three critical quantum control tasks relevant to quantum information processing. First, we addressed coherent quantum population transfer in multi-level $\Lambda$ systems, describing a variety of quantum systems and of relevance to quantum chemistry and solid state physics, where we achieve high-fidelity population transfer in spite of dissipation and cross-talk. Second, we optimise Rydberg gates in neutral atom quantum devices, focusing on enhancing gate fidelities and robustness to time-dependent noise, which is crucial for scalable quantum computing. Third, we developed efficient reset protocols for superconducting transmon qubits under realistic experimental constraints like bandwidth limitations, essential for fast quantum circuit execution. Here, we discover a novel, physically-feasible reset waveform which achieves an order of magnitude higher reset fidelity than any previous work. Fig. 1 demonstrates the efficacy of our proposed method in finding higher-fidelity solutions while reducing computational demand.

| Method | $\mathcal{F}_\pi$ | Exp. Feasible |
|---|---|---|
| Optimal Control (Giannelli et al., 2022b)[4] | $\overline{0.890} \pm 0.064$ | No |
| Analytic (Vasilev et al., 2009) | 0.901 | Yes |
| RL (Giannelli et al., 2022a) | $\overline{0.930} \pm 0.034$ | No |
| RL (Norambuena et al., 2023)[5] | 0.83 | Yes |
| RL (this work) | $\mathbf{\overline{0.999} \pm 0.0003}$ | Yes |

Table 2: We benchmark different methods for optimising coherent quantum population transfer in a multilevel $\Lambda$ systems by optimising $\mathcal{F}_\pi$ for a complete ground state rotation. Averaged over 32 random seeds, our method achieves significantly higher fidelity than prior work with reduced sensitivity to the initial seed, while yielding experimentally feasible control signals.

### 5.0.1 EXPERIMENTAL IMPLEMENTATION

**Constrained RL Implementation.** To enforce the constraint $N_{\text{Sim}}(a) < N_{\text{Sim}}^{\text{max}}$ on actions $a$ sampled by optimal policies $\pi$ in the bandit setting, we modify the reward function by assigning a penalty reward $r_{\text{penalty}}$. In the bandit setting, $r_{\text{penalty}}$ is assigned to any policy where $N_{\text{Sim}}(a) >= N_{\text{Sim}}^{\text{max}}$, and the value is chosen to be lower than any other possible reward in the environment, ensuring that the optimal policy cannot include states violating the constraint (Altman, 2021). This approach can be easily extended to multi-step settings when the bounds of the reward function are known, which is the case here (Altman, 2021). The final reward function is then defined as

$$L = \begin{cases} r_{\text{penalty}} & \text{if } N_{\text{Sim}}(a) >= N_{\text{Sim}}^{\text{max}} \\ L_1 & \text{else} \end{cases} \tag{7}$$

where $L_1$ is defined in equation 6.

**Additional Implementation Details.** We leverage the Qiskit-Dynamics Solver interface (Puzzuoli et al., 2023) for constructing both Hamiltonians and collapse operators, enabling the simulation of open quantum systems through the dissipative Gorini-Kossakowski-Sudarshan-Lindblad equations. We employ the Diffrax ODE solver (Kidger, 2022) for quantum system simulation, which utilise adaptive step-sizing techniques to efficiently integrate the first-order linear differential equations and PureJAXRL for implementing PPO algorithms (Lu et al., 2022).

### 5.1 COHERENT QUANTUM POPULATION TRANSFER IN MULTI-LEVEL ELECTRONIC SYSTEMS

Controlling the quantum dynamics of multilevel systems is ubiquitous for quantum information processing and is also relevant for solid state physics and chemistry (Bergmann et al., 2019; Vitanov et al., 2017). We focus on a common experimental setup (Vitanov et al., 2017), also known as a $\Lambda$ system, where two time-dependent control signals with amplitudes $\Omega_S$, $\Omega_P$ couple two electronic states with relative time-dependent frequency detunings $\Delta_P$ and $\Delta_\delta$ (cf. App. Sec. B for more details). These four parameters consist the control fields defined in equation 2. Many analytically optimal pulses exist for idealised and isolated three level systems (Kuklinski et al., 1989; Vasilev et al., 2009). We include dissipation, parametrised by rate $\Gamma$, as well as an additional excited state detuned positively $\Delta_X$ from the excited state addressed with $\Omega_{S/P}$ to which cross talk must be suppressed (cf. App. Sec. B for details). This represents a common physical configuration describing, for instance, nitrogen vacancy centres (Balasubramanian et al., 2009), quantum dots (Economou et al., 2012), circuit-QED systems (Novikov et al., 2015), or single atoms (Ernst et al., 2023). We present and benchmark results on optimising population transfer from one ground state $|g_1\rangle$ to another $|g_2\rangle$. We fix $\Gamma = 1$, $\Omega_{max} = 30$ and $\Delta_X = 100$.

We observe in Tab. 2 that the fidelities $\mathcal{F}_\pi$ achieved in a 4-level $\Lambda$ system are significantly higher than state of the art and also more robust across different random starting points, highlighting the superiority of RL over methods which directly differentiate the control action with respect to the fidelity. We further find that the learned pulses are physically viable, while prior work (Giannelli et al., 2022b;a) found infeasible solutions, which exhibit non-zero amplitudes at the start or end or have instantaneous parameter changes which cannot be realised on bandwidth limited hardware.

---

[4]Direct Differentiation of Signal with BFGS (Fletcher, 1987) with max iterations of 10000.

[5]No code or further data were available to benchmark this in our environment.

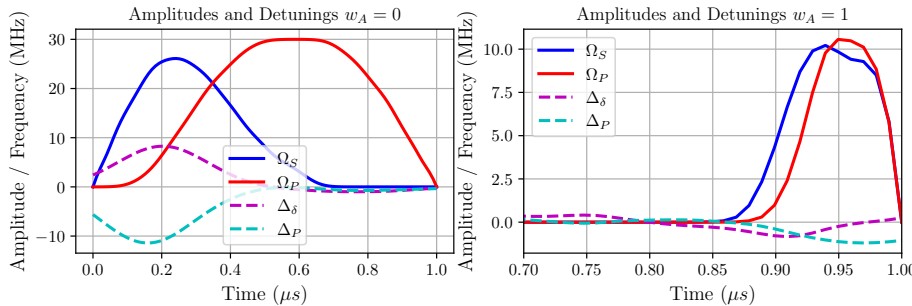

Figure 3: Shown are example control signals generated for different pulse area penalties. For $w_A = 0$ (left), the agent seeks to maximise $\Omega$ at all times after a fast rise and compensates cross-talk with frequency chirping. For $w_A = 1$ (right), we plot only the time interval $[0.7, 1]$, as the pulse amplitudes are zero otherwise and show that the agent discovers pulses which reminisce of two interleaved Gaussians, but exhibit non zero two-photon detuning $\Delta_\delta = 0$ to cancel cross-talk (cf. App. Sec. B), which differs from the original proposal for coherently transferring population between two groundstates (Kuklinski et al., 1989).

Sweeping $w_A$ cf. equation 7 we find particular signals which have pulse areas which approach those quoted in (Norambuena et al., 2023) (cf. Fig. 7 in the App.). Example signals differ significantly for different pulse area penalties which is shown in Fig. 3.

Random fluctuations or noise of either signal $\Omega_{S/P}$ or $\Delta_{\delta/P}$ are not as detrimental to the overall fidelity. We implement an Ornstein–Uhlenbeck noise process for both $\Delta_{\delta/P}$ and $\Omega_{S/P}$, a Markovian noise model which creates continuous noise $\nu_t$ in time with mean $\mu$ and standard deviation $\sigma$ (for details cf. App. Sec. E.3). Such noise typically arises from a variety of imperfections in the signal chain, as well as quantum system level noise, such as magnetic field fluctuation or motion. Using unbiased ($\mu = 0$) noise with various standard deviations exemplifies good robustness to low noise levels as shown in Fig 8 (cf. App. Sec. B) where we attain $> 0.99$ mean fidelity for $\sigma_\Omega = \sigma_\Delta = 0.1$. Further increasing $\sigma$ leads to significantly reduced population transfer fidelities which we address with multi-step RL in Sec. 5.4. Solutions for a larger variety of system parameters and an extension to partial state transfer are shown in App. Sec. B.

## 5.2 RYDBERG GATES

Neutral atom quantum devices have shown promise for realising scalable, logical quantum computing (Bluvstein et al., 2023). The realisation of quantum computing requires a two-qubit gate (Nielsen & Chuang, 2010) which relies on the interaction of multiple atomic qubits which are brought in relative proximity (a detailed description of the Hamiltonian is provided in App. Sec. C) and addressed with laser beams. We consider an optimisation of the Rydberg gate (Lukin et al., 2001) under realistic experimental conditions. We include finite Blockade strength, as well as signal perturbations in amplitude and frequency.

We consider the most widespread implementation of a Rydberg $C$-$Z$ gate (a single photon Rydberg gate (Levine et al., 2019a; Jandura & Pupillo, 2022)) with a single pulse of amplitude $\Omega_P$ and time-dependent frequency $\Delta_P$ which has known solutions. This is compared to the two-photon Rydberg $C$-$Z$ gate which uses two time-dependent signals with amplitudes $\Omega_P, \Omega_S$ and frequencies $\Delta_P, \Delta_S$ (akin to the $\Lambda$ system). The single photon Rydberg gate is extremely vulnerable to time-dependent noise as shown in App. Fig. 11. This motivates the determination of an optimal pulse sequence for the two-photon Rydberg gate which exhibits superior robustness by an order of magnitude. Finding optimal protocols which simultaneously optimise both amplitude and frequency of Pump and Stokes beams is extremely challenging since the Hilbert space is over 10- dimensional. Compared to Saffman et al. (2020) we find a solution (cf. App. Fig. 10) which is higher fidelity $\mathcal{F} = 0.9993$ than their analytic solution $\mathcal{F} = 0.99$, as well as their numerical solution $\mathcal{F} = 0.997$ and faster $0.25\mu s$ compared to their $1\mu s$ numerical solution. Compared to Sun (2023) we achieve similar fidelities but with an order of magnitude lower peak Rabi frequencies which implies lower laser power requirements. Moreover, we implement a direct C-Z gate which does not require any ad-

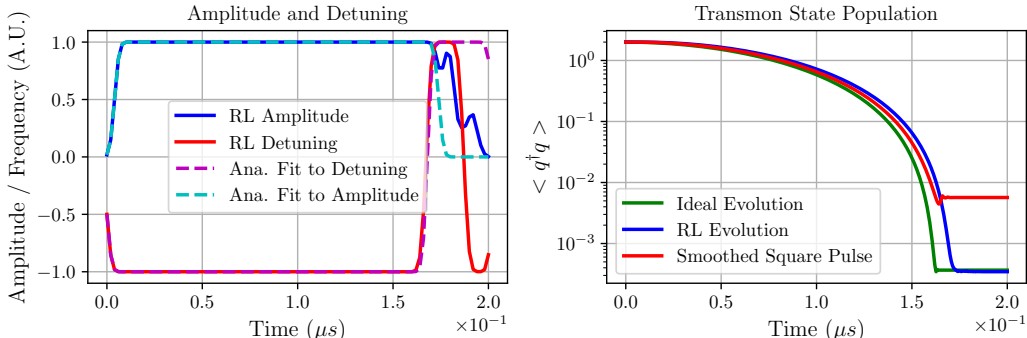

Figure 4: Optimal Waveform for Transmon Reset (left) discovered by RL and corresponding state evolution (right). The RL waveform (solid lines) amplitude evolution reminisces of a square-top Gaussian, with a smooth Heaviside-detuning that accounts for time dependent frequency shifts. Equivalent reset performance is found by fitting a Heaviside-detuning reset and a Gaussian square amplitude waveform (dashed lines), simplifying experimental calibration. Our approach shows reset errors of 0.03% matching the performance under an experimentally unrealistic ideal square pulse, and showing an order of magnitude improvement over a smoothed square pulse.

ditional ground state rotations. We directly differentiated the input action with respect to the fidelity with a BFGS (Fletcher, 1987) method over 1000 iterations and for 32 random seeds and achieved a mean fidelity of $0.914 \pm 0.0742$ (one s.d.) showing the superiority of RL in achieving robust, high fidelity solutions as alluded to in Tab. 1. The enhanced computational scaleability offered by our implementation could be used to optimise higher order gates like a $C^k$-$Z$ which are also robust.

### 5.3 TRANSMON RESET

Superconducting quantum bits (qubits) have played a central role in quantum computing break-throughs, including the demonstration of quantum supremacy (Arute et al., 2019) as well as the suppression of errors with the surface code (Acharya et al., 2023). The transmon Koch et al. (2007), a widely used superconducting qubit, operates within its two lowest energy levels to form a qubit subspace. Recent advances have extended transmon lifetimes beyond 0.5 ms (Wang et al., 2022), enabling longer quantum circuits and the implementation of error correction codes. To maximise circuit operations within the qubit's lifetime, transmons must be reset efficiently with high fidelity.

Two main reset techniques exist: conditional reset (Ristè et al., 2012), which follows state measurement, and unconditional reset (Magnard et al., 2018), which is faster and more robust. We focus on optimising waveforms for unconditional reset (cf. App. Sec. D). The reset rate is proportional to drive strength, theoretically favouring high-amplitude square pulses for maximum fidelity. However, a drive-induced Stark shift alters the transmon's resonance frequencies Zeytinoğlu et al. (2015). In ideal conditions, a square pulse with a calibrated frequency can counter this shift. IBM demonstrated this approach experimentally, achieving 0.983 fidelity, while simulations under ideal conditions reached 0.996 fidelity (Egger et al., 2018a). This mismatch could be explained by experimentally realistic bandwidth constraints as square pulses have a finite rise and fall time, which induces a time dependent frequency shift. While optimal control Gautier et al. (2024) has been applied to the task of reset pulse optimisation, minimal bandwidth constraints implied that no novel waveforms were found for improving the reset transition in realistic experiments. Using BFGS with direct differentiation of the input signal failed to optimise multi-objective reward functions or satisfy realistic signal constraints. When optimising solely for fidelity, it remained slow and prone to local minima due to the large search space of non-smooth actions.

We apply scaleable RL to optimise the transmon reset waveform under bandwidth constraints imposed by Gaussian-smoothing (for further details cf. App. Sec. E.2). Considering state of the art parameters, as given in the IBMQ experiment – a qubit lifetime $T_1$ of $500\mu$s – we find that our RL approach achieves 0.9997 fidelity under realistic bandwidth constraints shown in Fig. 4 (cf. App. D for further implementation details). This is compared with a perfect square pulse - which is not ex-

perimentally realistic - without any smoothing, and a calibrated square pulse with smoothing - which represents prior work (Egger et al., 2018a). The RL waveform matches the theoretical optimal fidelity of the perfect square pulse and improves the fidelity of waveform used in prior work Egger et al. (2018a) by an order of magnitude. In App. Sec. D we explicitly compare the results with the parameters used in (Egger et al., 2018a), and find that the RL discovered reset waveform achieves the fidelity 0.997 of the ideal square pulse compared to the measured fidelity of 0.983. A fitted Heaviside detuning function from the RL-discovered waveform corrects the drive-induced Stark shift, simplifying experimental calibration, which we dub Heaviside-Corrected Gaussian Square (HCGS) and explain further in App. Sec. D.1. Further results and extensions are provided in App. Sec. D.

### 5.4 Multi-Step Reinforcement Learning

We study the effectiveness of multi-step reinforcement learning (RL) strategies in achieving high fidelity control solutions under adverse noise conditions. Feedback on nanosecond timescales has been demonstrated experimentally (Álvarez et al., 2022; Koch et al., 2010), supporting this approach. This feedback can be realised by measuring classical signal noise without affecting quantum coherence. For example, in atomic quantum systems, laser intensity $I$ can be measured at an arm separate from the quantum system, as $\Omega \propto I$. Changes in $I$ directly modulate $\Omega_{S/P}$ and thereby provide feedback for multi-step learning.

In multi-step RL, the agent aims to maximise cumulative rewards over multiple steps, unlike the bandit setting where actions are independent. In our setup, at the start of each episode, a parameter $\mu$ is sampled uniformly from $[-\sigma_{\max}, \sigma_{\max}]$ to initialise an Ornstein–Uhlenbeck noise process (see App. Sec. E.3 equation 25). The agent's control signal $a_t = \Omega_i(t)$ (amplitudes only) is affected by this noise, resulting in $\Omega'_i = \Omega_i + \nu_t$. In bandit RL, the agent does not observe the noise $\nu_t$ and selects the action in one step. Conversely, in multi-step RL, each episode is divided into four sections of 8 action samples corresponding to $0.25\mu s$ each. The agent initially observes $O_t = 0$ but receives the value of $\mu$ at times $t = 0.25, 0.5$, and $0.75\mu s$ (further implementation details are given in App. Sec. A.2. App. Fig. 9 illustrates that multi-step RL outperforms the bandit approach, especially as $\mu$ increases beyond 10.

## 6 Conclusion

In this work, we introduced a novel reinforcement learning implementation for controlling open quantum systems by formulating quantum control as a constrained RL problem. By integrating physics-based constraints that exclude control signals inducing overly fast quantum dynamics and enforcing smooth pulses with finite rise-time, we enhanced both the quality of control solutions and computational scalability. Our approach outperformed existing methods on three key quantum control tasks, achieving higher fidelities and increased robustness to time-dependent noise. We wish to highlight here, that especially for the Transmon qubit we find novel waveforms that can be described with smooth functional parametrisation and realised with off the standard hardware. We are actively working on verifying the quality of our found solutions on physical devices. For future work, we envision extending our implementation to more complex quantum systems, this includes multi-qubit systems and higher-dimensional state spaces. Additionally, future work would extend this to quantum control tasks which require multiple sequential quantum gates or other concatenated control operations. Exploring adaptive constraint mechanisms that adjust during the learning process could further improve performance. Additionally, incorporating more advanced and physically relevant noise models and collaborating with experimental physicists to validate our control policies on actual quantum hardware would accelerate the practical development of quantum technologies.

**Limitations.** While our physics-informed constrained RL implementation enhances computational efficiency and solution quality, it may limit the exploration of control strategies that involve very fast and non-adiabatic quantum dynamics. The method's effectiveness also relies on accurate modelling of quantum systems, so models would first have to be established for black box systems or more complicated real world devices. Although we address certain types of noise and perturbations, fully accounting for all experimental imperfections is an area for future work and we could consider sampling from real devices.

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

## SUPPLEMENTARY MATERIAL

Here we present detailed explanations of the extended RL background, quantum dynamical systems simulated in the main paper, show auxiliary results and explain our implementation in greater detail.

## A  RL BACKGROUND

### A.1  BANDIT SETTING IN REINFORCEMENT LEARNING

In the bandit setting, the RL problem is simplified as there is no state transition, only actions and rewards. Each action $a \in \mathcal{A}$, which are time dependent quantum control signals $\Delta_i, \Omega_i$ yields a reward from a stationary probability distribution. The objective is to maximise the expected reward over a sequence of actions.

Formally, given a set of actions $\mathcal{A}$, each action $a \in \mathcal{A}$ has an unknown reward distribution with expected reward $R(a)$. The goal is to find the action $a^*$ that maximises the expected reward:

$$a^* = \arg\max_{a \in \mathcal{A}} \mathbb{E}\left[R(a)\right] \tag{8}$$

This setting forms the basis for more complex RL problems.

### A.2  EXTENDED TIME HORIZON IN MULTI-STEP RL

For multi-step RL, we consider an extended time horizon. In contrast to the bandit setting, each episode is divided into four sections, each of length 8 action samples. The agent does not observe any information about the noise at time step $t = 0$, with the observation $O_t = 0$. However, at time steps $t = 0.25, 0.5$, and $0.75$, the agent receives the value of mean noise $\mu$ sampled at the beginning of the episode. Formally, the observation function $\mathcal{O}_t$ is defined as:

$$\mathcal{O}_t = \begin{cases} 0 & \text{if } t = 0 \\ \mu & \text{if } t = 0.25k \text{ for } k = 1, 2, 3 \end{cases} \tag{9}$$

The agent's policy $\pi$ then uses this observation to decide the action at each time step, where $\tilde{S}_t$ is the union of the state in the bandit setting $s_t = \rho_t$ and the observation $\mathcal{O}_t$ which defines an action $a_t$ through a conditional probability distribution $\mathcal{P}$:

$$\pi(\tilde{s}_t, a_t) = \mathcal{P}[a_t | \tilde{s}_t, \theta] \tag{10}$$

In general extended time horizon RL, the agent must consider the long-term consequences of its actions. This is formalised through the discount factor $\gamma$, which ensures that future rewards are appropriately weighted. Given that we have a fixed number of four steps we set the discount factor to zero.

### A.3  PROXIMAL POLICY OPTIMISATION (PPO)

Proximal Policy Optimisation (PPO) is a popular algorithm in modern RL, combining the benefits of policy gradient methods with stability improvements. PPO aims to optimise the policy by ensuring that updates do not deviate too much from the previous policy. This is achieved using a clipped objective function.

The objective function in PPO is defined as:

$$L^{\text{CLIP}}(\theta) = \mathbb{E}_t\left[\min\left(r_t(\theta)\hat{A}_t, \text{clip}(r_t(\theta), 1 - \epsilon, 1 + \epsilon)\hat{A}_t\right)\right] \tag{11}$$

where:

- $r_t(\theta) = \frac{\pi_\theta(a_t | s_t)}{\pi_{\theta_{\text{old}}}(a_t | s_t)}$ is the probability ratio under the new and old policies.

- $\hat{A}_t$ is an estimate of the advantage function at timestep $t$.

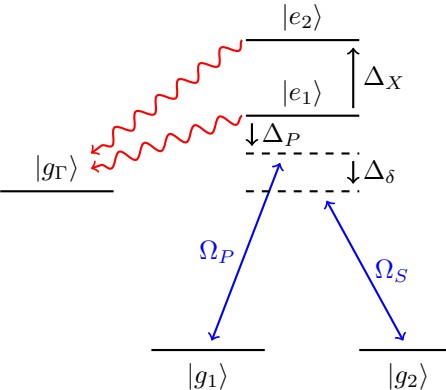

Figure 5: Energy level diagram for four level Λ system with state $|e_2\rangle$, detuned positively by $\Delta_X$ from $|e_1\rangle$, to which cross talk is supressed. There is an additional state $|g_\Gamma\rangle$ which does not partake in the unitary dynamics, but to which the excited states decay (cf. red dotted lines). This gives rise to a lower bound in attained population transfer fidelities. The laser couplings from Stokes and Pump laser are shown in blue.

- $\epsilon$ is a hyperparameter that controls the clipping range.

The clipping mechanism in the objective function ensures that the new policy does not deviate significantly from the old policy, thereby improving training stability and preventing large, destabilising updates.

PPO also incorporates an entropy bonus to encourage exploration and prevent premature convergence to suboptimal policies. The overall objective with the entropy bonus can be written as:

$$L(\theta) = \mathbb{E}_t \left[ L^{\mathrm{CLIP}}(\theta) + c_1 \hat{A}_t + c_2 E[\pi_\theta](s_t) \right] \tag{12}$$

where $c_1$ and $c_2$ are coefficients, and $E[\pi_\theta](s_t)$ denotes the entropy of the policy at state $s_t$.

In summary, PPO effectively balances exploration and exploitation while ensuring stable policy updates, making it a robust choice for RL in quantum control tasks.

## B   ELECTRONIC Λ SYSTEMS

A very common system configuration in quantum information contains two ground-states $|g_1\rangle$, $|g_2\rangle$, coupled by a common excited state $|e_1\rangle$, as is required for the implementation of many quantum population transfer protocols, such as **St**imulated **R**aman **A**diabatic **P**assage (STIRAP) (Vitanov et al., 2017). We also include an additional excited state $|e_2\rangle$, detuned positively by an amount $\Delta_X$ from $|e_1\rangle$ to show the effect of crosstalk due to a coupling to an undesired transition. This configuration is ubiquitous and arises naturally in colour centres, quantum dots or other electronic quantum systems. An explicit energy level diagram is provided in Fig. 5. $\Omega_{P/S}$ denote the Rabi frequencies of the Pump and Stokes pulses respectively and $\Delta_{P/\delta}$ are the detuning of the Pump pulse from resonance as well as the two photon detuning respectively. The Hamiltonian $H_\Lambda$ used to model the unitary dynamics, defined in the basis $(|g_1\rangle, |g_2\rangle, |e_1\rangle, |e_2\rangle)$, after an application of the rotating wave approximation reads:

$$H_\Lambda/\hbar = \begin{bmatrix} 0 & 0 & \frac{\Omega_P}{2} & \frac{\Omega_P}{2} \\ 0 & \Delta_P - \Delta_\delta & \frac{\Omega_S}{2} & -\frac{\Omega_S}{2} \\ \frac{\Omega_P}{2} & \frac{\Omega_S}{2} & \Delta_P & 0 \\ \frac{\Omega_P}{2} & -\frac{\Omega_S}{2} & 0 & \Delta_P + \Delta_X \end{bmatrix} \tag{13}$$

All Rabi frequencies $\Omega_{P/S}$ are real. Additionally we include a sink state to which spontaneous emission occurs which couples equally to both excited state with rate $\Gamma/\sqrt{2}$, this is realistic insofar as spontaneous emission can always occur to states outside the manifold of interest, but as we do

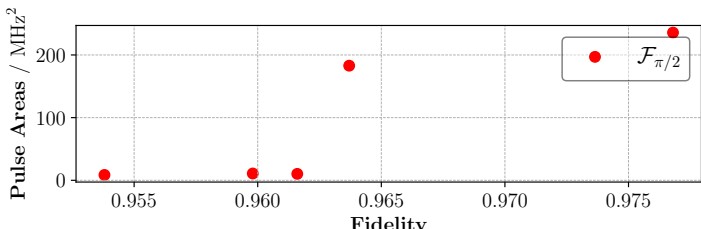

Figure 6: We show pulse area versus the fidelity for a partial state rotation $\mathcal{F}_{\pi/2}$ and sweep $w_A$, the pulse area penalty weight defined in equation 7 (cf. App. Sec. E.2 for more details) over a range of values $[0, 0.1, 0.25, 1, 2]$ and approach minimal pulse area with respect to Ref. (Norambuena et al., 2023) whilst achieving significantly higher fidelities $> 0.975$. There is a clear trade off and lower pulse areas generally adversely affect fidelities.

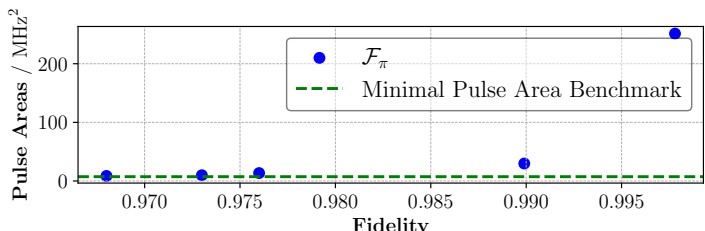

Figure 7: We sweep $w_A$, the pulse area penalty weight (cf. equation 7) over a range of values $[0, 0.1, 0.25, 1, 2]$ and approach minimal pulse area with respect to Norambuena et al. (2023) (green dotted line) whilst achieving significantly higher fidelities $\mathcal{F}_\pi > 0.83$ (cf. Norambuena et al. (2023)). There is a clear trade off and lower pulse areas generally adversely affect fidelities.

not consider spontaneous emission to $g_1$ or $g_2$ we obtain lower bounds on any population transfer fidelities $\mathcal{F}$. The Lindbladian operator reads; $\Gamma/\sqrt{2} \, |g_\Lambda\rangle \langle e_i|$. In the main text, the initial state is always fixed as $|g_1\rangle$, but the desired final states are $|g_2\rangle$, as well as $|+\rangle = 1/\sqrt{2}(|g_1\rangle + |g_2\rangle)$, such that we have two fidelity measures, $\mathcal{F}_\pi$ and $\mathcal{F}_{\pi/2}$ where the subscript denotes the rotation angle in the ground state basis. Generally, the protocol can be extended to arbitrary angles $\theta$, but we focus on two without loss of generality. $\theta = \pi$ is an extremely common scenario which is described extensively in the literature (Vitanov et al., 2017) and $\theta = \pi/2$ is also common and has been described in Ref. (Vitanov et al., 1999).

For the $\Lambda$ system we introduce an additional reward term which reads $-w_x \cdot (\langle e_1\rangle + \langle e_2\rangle)$. This assigns lower rewards to non-coherent dynamics, since we seek coherent population transfer and speeds up the learning dynamics.

We showcase two particular reference pulses for different pulse areas in Fig. 3. Trade-offs between pulse areas and population transfer fidelity are shown in Figs. 6 and 7 for $\theta = \pi/2, \pi$ respectively and we show that we approach the lower pulse area limit described in Ref. Norambuena et al. (2023). We also show robustness to time dependent noise in Fig. 8.

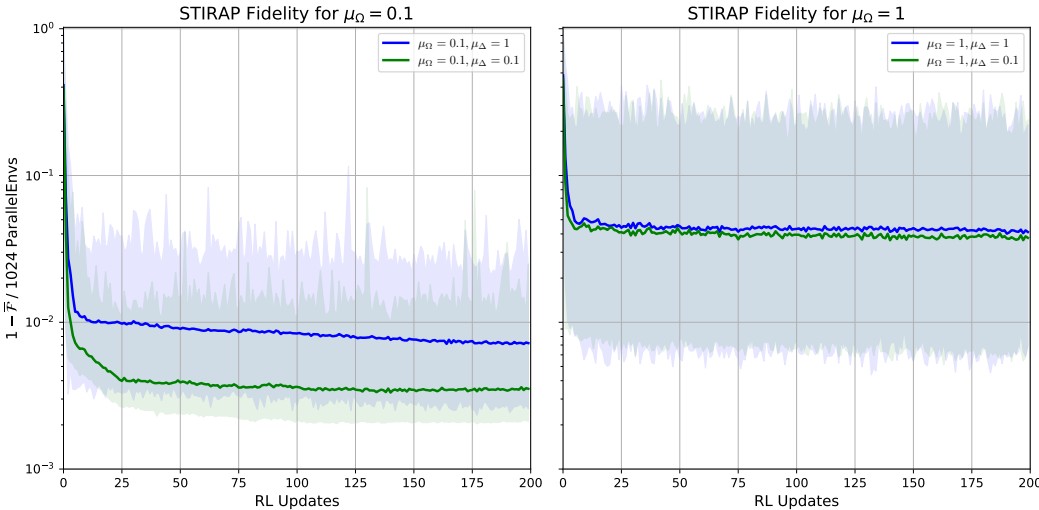

Figure 8: Robustness of protocol to randomly generated noise with $\mu_\Omega = 0.1$ MHz (left) and $\mu_\Omega = 1$ MHz (right) plotted on a logarithmic scale and averaged over multiple seeds. The solid lines show the average fidelity, while the shaded regions indicated min/max fidelity over all parallel environments. For small noise levels $F > 0.99$ as shown in the left plot for $\mu_\Delta = 0.1$ MHz, but as it increases fidelities drop to just below 0.97. $N_{\max}$ is chosen such that even with noise all parallelised runs can be solved for $\rho(t)$.

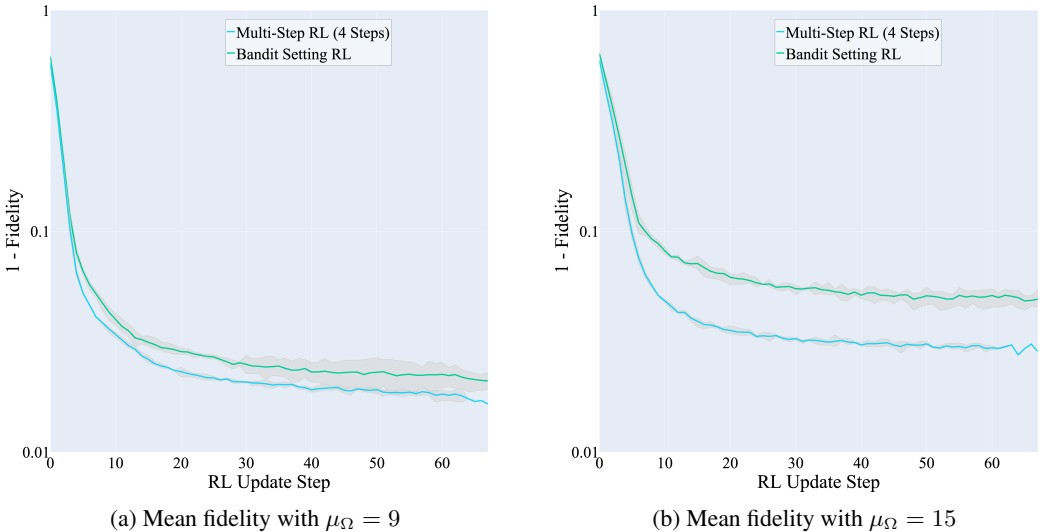

(a) Mean fidelity with $\mu_\Omega = 9$          (b) Mean fidelity with $\mu_\Omega = 15$

Figure 9: Comparison of mean infidelity for different values of $\mu_\Omega$ for bandit RL and multi-step RL. We observed several percent reduction in infidelity for larger noise bias $\mu_\sigma$ by using multi-step RL over the bandit setting.

## C    RYDBERG GATES

We first consider a Rydberg gate based on a single laser excitation which is near resonant with the ground-state qubit $|1\rangle$ and Rydberg level $|r\rangle$ transitions. Following the implementation experimentally shown in (Levine et al., 2019b) and the Hamiltonian definition given in (Pagano et al., 2022)

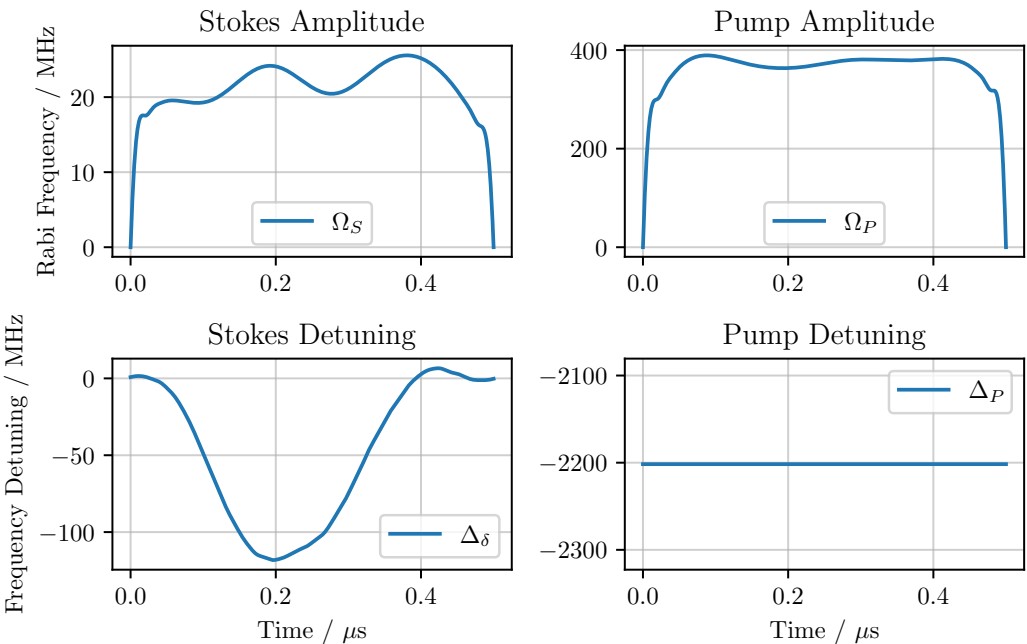

Figure 10: We show optimal signals for a two photon Rydberg gate directly realising a C-Z gate, with amplitudes (i.e. Rabi frequencies) for Stokes and Pump pulses in MHz shown in the top column. The effective maximum Rabi frequency of the pump pulse ($\Omega_P^2/2 * \Delta_P$) is $\approx 20$ to match that of the Stokes pulse. Detunings of the Stokes and Pump pulse are shown in the bottom row. Note the symmetry of the Stokes detuning in time which shows a semblance of a reflection symmetry about its centre which ensures that a relative $\pi$ phase is acquired between the basis states (cf. equation 17) and their populations largely return to their initial values. This pulse yields a fidelity of $0.9987$ for a $0.5\mu s$ duration and can be shortened to $0.25\mu s$ with all signals re-scaled by 2 which yields a fidelity of $0.9993$ since we are mainly Rydberg level lifetime limited, with a finite blockade strength of $500$MHz. This pulse is also shown to exhibit very little variation across different blockade strengths.

the Hamiltonian for the one-photon Rydberg gate $H_{r_1} = H_0 + H_{\text{int}}$ reads:

$$\frac{H_0}{\hbar} = \sum_i^2 \left[ \frac{\Omega(t)}{2} (|r\rangle \langle 1|_i + |1\rangle \langle r|_i) - \Delta(t) |r\rangle \langle r|_i \right]$$

$$\frac{H_{\text{int}}}{\hbar} = B |r, r\rangle \langle r, r| \tag{14}$$

Here $\Omega(t)$ and $\Delta(t)$ are real amplitudes and detunings of a Rydberg laser and $B$ describes the dipole blockade strength. The Linbladian terms are described by the addition of a sink state $g_\Gamma$ which imposes a lower bound on fidelity since any population which spontaneously decays leaves the computational subspace, as for the $\Lambda$ system. They read; $\sum_i \Gamma_r (|g_\Gamma\rangle (\langle r, i| + \langle i, r|) + \Gamma_r (|g_\Gamma\rangle \langle r, r|)$, where $\Gamma_r$ describes the decay rate of the Rydberg level. Many optimisation protocols consider $B \to \infty$, since the Rydberg gate operates in the regime $\Omega << B$ which precludes coupling of both qubits to $|r\rangle$, however we fix $B$ to a finite but realistic value in the range of hundreds of MHz (Pagano et al., 2022; Pelegrí et al., 2022; Sun, 2023).

One of the drawbacks of this implementation, as described in the main test however, is that it is not particularly robust in the face of signal imperfections and noise. Using the physics of a two photon process (similar to the $\Lambda$ system dynamics) we follow the Hamiltonian definition $H_{r_2} = H_{0,2} + H_{\text{int},2}$ for a two-photon Rydberg gate given in (Sun, 2023) (where H.C. denotes the hermitian conjugate):

$$\frac{H_{0,2}}{\hbar} = \frac{\Omega_P(t)}{2} |10\rangle \langle e0| + \frac{\Omega_S(t)}{2} |e0\rangle |r0\rangle + \text{H.C.} + \Delta_P(t) |e0\rangle \langle e0| + \Delta_S(t) |r0\rangle \langle r0|, \tag{15}$$

with time-dependent Rabi frequencies $\Omega_P(t), \Omega_S(t)$, and values for the one photon detuning $\Delta_P$ and two-photon detuning $\Delta_S$. The Hamiltonian terms for $|01\rangle$ follow analogously from symmetry considerations by swapping all qubits in their respective state in $H_{0,2}$.

The interaction Hamiltonian $H_{int,2}$ for the state $|1,1\rangle$ consists of the atom light interaction as well as the dipole-dipole interaction akin to equation 14. A basis transformation simplifies the Hamiltonian, the new basis states read $|\tilde{e}\rangle = (|e1\rangle + |1e\rangle)/\sqrt{2}, |\tilde{r}\rangle = (|r1\rangle + |1r\rangle)/\sqrt{2}$ and $|\tilde{R}\rangle = (|re\rangle + |er\rangle)/\sqrt{2}$, after the rotating wave approximation, and effectively neglecting $|ee\rangle$, as we are in the regime where $\Delta_P >> \Delta_S$, $H_{int,2}/\hbar$ can be expressed as:

$$
\begin{aligned}
\frac{H_{int,2}}{\hbar} &= \frac{\sqrt{2}\Omega_P(t)}{2}|11\rangle\langle\tilde{e}| + \frac{\Omega_S(t)}{2}|\tilde{e}\rangle\langle\tilde{r}| + \frac{\Omega_P(t)}{2}|\tilde{r}\rangle\langle\tilde{R}| + \frac{\sqrt{2}\Omega_S(t)}{2}|\tilde{R}\rangle\langle rr| + \text{H.C.} \\
&+ \Delta_P(t)|\tilde{e}\rangle\langle\tilde{e}| + \Delta_S(t)|\tilde{r}\rangle\langle\tilde{r}| + (\Delta_P(t) + \Delta_S(t))|\tilde{R}\rangle\langle\tilde{R}| + 2\Delta_P(t)|rr\rangle\langle rr| \\
&+ B|rr\rangle\langle rr|
\end{aligned}
\tag{16}
$$

Parameters $\Omega_{S/P}, \Delta_{S/P}, B$ are defined as in equation 14. The Linbladian decay terms for the two photon Rydberg gate are described similarly as for the one photon Rydberg gate. They read; $\sum_i \Gamma_r(|g_\Gamma\rangle)(\langle r,i| + \langle i,r|) + \Gamma_r(|g_\Gamma\rangle\langle r,r|) + \sum_i \Gamma_e(|g_\Gamma\rangle(\langle e,i| + \langle i,e|) + \Gamma_e(|g_\Gamma\rangle\langle e,e|)$, where $\Gamma_r$ describes the decay rate of the Rydberg level and $\Gamma_e$ the decay of the excited level $|e\rangle$ where for typical atoms $\Gamma_e >> \Gamma_r$.

Akin to the $\Lambda$ system we introduce an additional reward term which reads $-w_x \cdot (\langle rr\rangle + \langle\tilde{e}\tilde{e}\rangle + \langle\tilde{r}\tilde{r}\rangle + \langle rr\rangle + \langle\tilde{R}\tilde{R}\rangle)$. This assigns lower rewards to non-coherent dynamics, since we seek coherent population transfer and speeds up the learning dynamics.

The fidelity $\mathcal{F}_R$ is defined by the Bell state fidelity as is common in optimisation protocols of the Rydberg gate (Jandura et al., 2023):

$$
\mathcal{F}_R = \frac{1}{16}|1 + \sum_{10,01,11} e^{-i\theta_q}\langle q\rangle\psi_q^0|^2,
\tag{17}
$$

without loss of generality, we focus on the C-Z gate where $\theta_q = 0$, except $\theta_{1,1} = \pi$, this is particularly useful insofar as it does not require additional single qubit rotations (in comparison to a general $C(\theta)$ gate) and does not introduce any further time overhead associated with additional rotations.

As described in the main text, we focus on the implementation of a two-photon Rydberg gate. For this, we fix the detuning of the pump pulse to a constant value, since a time-dependent frequency chirp offers no advantages in terms of achievable maximum fidelities, so we merely optimise its constant value. We fix $\Omega_S$ to a maximum value of 40 and $\Omega_P^2/(2\Delta_P)$ (the effective Rabi frequency) to a maximum value of 56.6 with a pump detuning of 2.5 GHz and obtain an optimal control signal which is shown in Fig. 10. It shall be noted note that the signals are different from results in the literature since we impose the realistic constraint of amplitudes to start and end at zero amplitude compared to (Sun, 2023). The optimal time-dependent control signals for a direct realisation of a C-Z gate are shown in Fig. 10. Following remarks made in Ref. (Sun, 2023) we show increased resilience to noise and achieve fidelities in excess of 0.99 even with significant levels of time-dependent noise, spontaneous emission (using realistic parameters for a $^{87}\text{Rb}$ (Sun, 2023) atom) and a finite blockade strength of 500 MHz as shown in Fig. 11.

## D  TRANSMON QUBIT RESET

Methods for unconditional transmon qubit reset with fixed-frequency devices involve using the coupling of a transmon to a low lifetime resonator through which excitations decay quickly. One particular hardware efficient protocol is based on a cavity-assisted raman transition utilising the drive-induced coupling between $|f0\rangle$ and $|g1\rangle$, where $|sn\rangle$ denotes the tensor product of a transmon in $|s\rangle$ and a readout resonator mode in the fock state $|n\rangle$. By driving the transmon simultaneously at the $|e0\rangle \leftrightarrow |f0\rangle$ transition and the $|f0\rangle \leftrightarrow |g1\rangle$ transition, we can form a $\Lambda$ system in the Jaynes-Cummings ladder which can be used to reset the transmon through fast single photon emission. The transmon reset Hamiltonian is given by

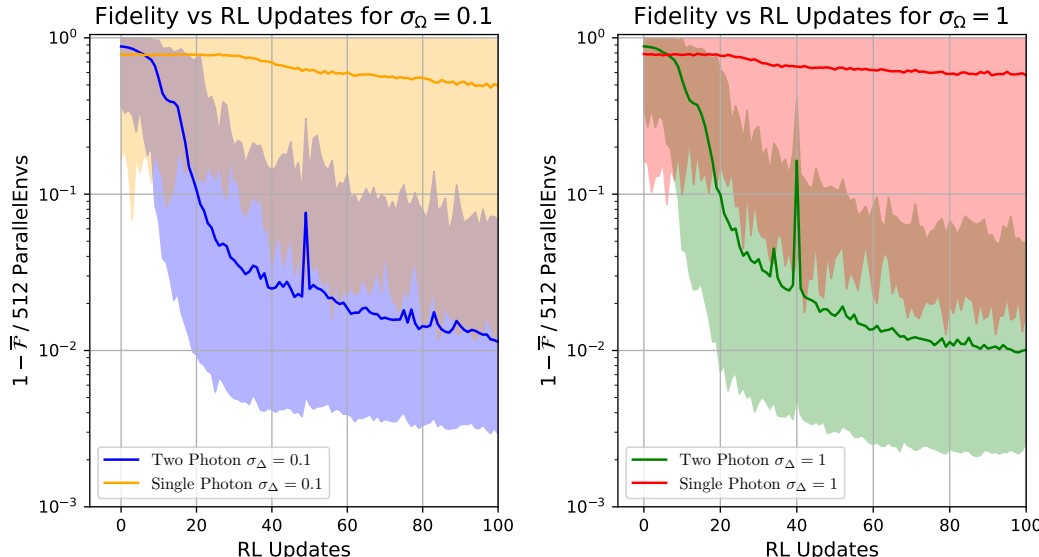

Figure 11: We explicitly compare the training of a single photon Rydberg gate (yellow and red) for moderate levels of amplitude and frequency noise $\sigma_\Delta = \sigma_\Omega = 0.1$ MHz (left) and $\sigma_\Delta = \sigma_\Omega = 1$ MHz, $\sigma_\Delta = \sigma_\Omega = 1$ MHz to a two photon Rydberg gate (blue and green) and show significantly superior resilience to time-dependent noise. The solid lines denote mean fidelity over 512 different random noise levels and the shaded lines denote the min/max noise levels. In the same number of RL updates, mean infidelity is about two orders of magnitudes lower for the single photon Rydberg gate which is the standard implementation of two qubit gates for Rydberg atoms. $N_{\max}$ is chosen such that even with noise all parallelised runs can be solved for $\rho(t)$.

$$\frac{H}{\hbar} = \chi a^\dagger a q^\dagger q + \frac{g\alpha}{\sqrt{2}\delta(\delta+\alpha)}\Omega(t)(q^\dagger q^\dagger a + a^\dagger qq) + (\Delta(t) + \delta_S(t))q^\dagger q \tag{18}$$

where $a(a^\dagger)$ is the resonator lowering (raising) operator, $q(q^\dagger)$ the transmon lowering (raising) operator, $\chi$ the transmon-resonator dispersive shift, $\alpha$ the transmon anharmonicity, $g$ the transmon-resonator coupling rate, $\delta$ the difference in the transmon and resonator resonant frequencies, $\Omega(t)$ the transmon drive amplitude, $\Delta(t)$ the transmon drive detuning, and $\delta_S(t)$ the drive-induced stark shift. As determined in Zeytinoğlu et al. (2015), this stark shift is to first order quadratic in the drive amplitude, $\delta_S(t) = k\Omega^2(t)$. For the transmon mode we consider three levels $|g, e, f\rangle$ coupled with a two level resonator. We neglect self-Kerr terms in the resonator mode as we target single photon populations where such non-linearities are not significant.

The Lindbladian for the transmon reset simulation is given by

$$\dot{\rho} = -i[H_S, \rho] + \kappa\mathcal{D}[\rho] + \Gamma\mathcal{D}[\rho] \tag{19}$$

with $\kappa$ describing the resonator decay rate, and $\Gamma$ the transmon decay rate.

We construct the transmon reset environment to match the physical parameters in Egger et al. (2018a), with maximum drive amplitudes of $330\,\mathrm{MHz}$, however with an additional small detuning control of up to $\pm 100$ kHz for frequency corrections. To represent bandwidth constraints, we add a Gaussian convolution of duration 14ns to the amplitude and detuning defined in equation 20. We use the same reward function as in previous environments with a calibrated max-steps limit of 900, and we neglect the pulse area penalty.

We first optimise the reset for a higher qubit lifetime of $T_1 = 500\,\mathrm{us}$, representing the transmon lifetimes currently attainable in experiment. Optimal waveforms and corresponding transmon populations are shown in Fig. 4, where the RL Pulse can achieve fidelities of 0.9997 even with realistic

bandwidth constraints. Notably, we find the RL agent consistently produces Gaussian-square like waveform for the drive amplitude, satisfying the high amplitude reset rate and optimising its smoothing. Novelty is observed in the time dependent detuning, which first stays at a constant frequency throughout the drive until at reset a quick shift is observed from negative to positive. This results in the overall waveform correcting dynamic stark-shifts induced by the drive amplitude fall time, allowing for near ideal reset fidelities.

When reducing the transmon lifetime to $T_1 = 48\mu$s as used in prior experimental work, the RL agent produces a similar waveform that achieves 0.997 fidelity matching the ideal calibrated square evolution, and achieving higher results than a calibrated square pulse which gets 0.992 and the experimental results in Egger et al. (2018a) which achieved 0.983. The success in optimising over a range of transmon $T_1$ lifetimes demonstrates that high fidelity unconditional reset can be achieved on current **N**oisy **I**ntermediate **S**cale **Q**uantum devices with advanced pulse control.

We further verify the RL solution quality in the context of a more significant Gaussian-smoothing kernel of 25ns and a qubit $T_1 = 500\mu$s, and find that it achieves high fidelities of 0.9995 while a standard square calibrated waveform deteriorates further to 0.9944 as errors arising from the uncorrected stark shifts become more significant.

### D.1 HEAVISIDE CORRECTED GAUSSIAN SQUARE

For the $|f0\rangle \leftrightarrow |g1\rangle$ transition in the reset process, the RL agent consistently finds a Gaussian Square pulse for the drive amplitude which reminisces of prior works, however with an additional Heaviside detuning profile as seen in Figure 4 which applies a frequency shift during the ring-down of the amplitude pulse.

This pulse, which we dub Heaviside-Corrected Gaussian Square (HCGS), directly corrects for a Hamiltonian which includes a drive-dependent stark-shift. Due to the finite ring-up time required for the amplitude, a negative frequency is applied to correct the positive amplitude-induced stark-shift. The negative frequency is applied throughout the reset until the ring-down. Before the ring-down of the square pulse, the Heaviside profile produces a positive detuning to correct for the negative amplitude-induced stark shift.

We note that this profile behaves quite similarly to past protocols such as DRAG where an additional phase component can be added to correct for unwanted Hamiltonian terms in the system. To further account for frequency bandwidth limitations, i.e. finite rise times for the phase control, the Gaussian Square duration $t_0$ and the Heaviside switch time $t_1$ can be at different points, with the Heaviside typically occurring a few nanoseconds earlier to account for the amplitude-driven stark shift.

Overall the HCGS reset pulse only requires 4 parameters, the amplitude $\Omega_0$ and duration $t_0$ of the Gaussian Square, along with the detuning magnitude $\Delta_0$ and the Heaviside switch time $t_1$. Since the calibration of the Gaussian square pulse parameters has already been described in various past works (Magnard et al., 2018; Egger et al., 2018a), to calibrate the HCGS reset only a further sweep of the detuning magnitude and switch time would be required to reach real world performance of RL-optimised waveforms.

## E IMPLEMENTATION DETAILS

### E.1 BENCHMARKING OF SIMULATION SPEED

Benchmarking absolute compute times across different hardware platforms, such as CPUs and GPUs, are challenging due to both systematic and random variations, even within the same architecture. Factors like GPU load balancing, data transfer overhead between the CPU and GPU, and kernel optimisations all influence performance, resulting in runtime fluctuations. Nonetheless, the speedups demonstrated in Fig. 12 highlight the advantages of GPU parallelisation for quantum simulations. We observe up to a two-order-of-magnitude improvement in speed per environment step, showcasing the significant performance benefits of running parallelised quantum simulations on GPUs, despite potential variability in the absolute timings.

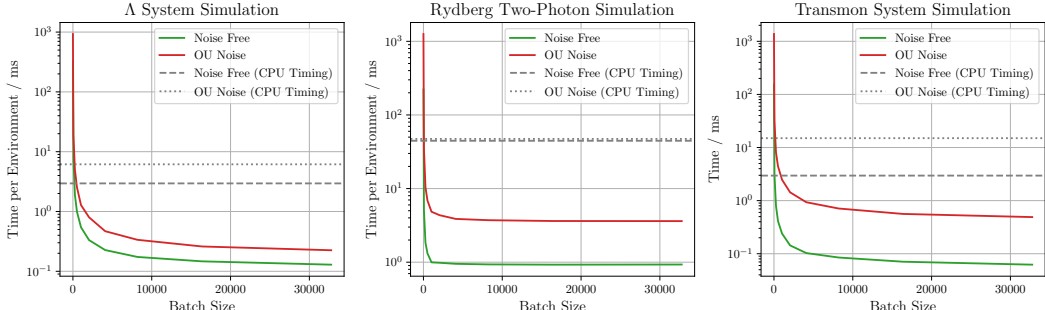

Figure 12: We compare the time per environment step for Qiskit Dynamics simulation across multiple environments ($\Lambda$ system, two photon Rydberg gate and Transmon) under noise-free and OU noise conditions. The left panel shows the $\Lambda$ system simulation timings, while the right panel illustrates the Rydberg two-photon simulation timings on a V-100 Nvidia GPU where we parallelise the simulation of several environments with different random actions and a fixed number of ODE solver steps= 4096. The solid lines represent the simulation times obtained with a GPU, while the dashed and dotted horizontal lines indicate the corresponding CPU timings (Apple Silicon M1) for Qiskit (noise-free and OU noise, respectively). Simulation time per environment is plotted on a logarithmic scale and in the best case we get up to about two orders of magnitude improvement in simulation time per environment in a larger batch by moving to a GPU.

### E.2 SIGNAL PROCESSING & ANALYSIS

The RL agent samples actions from the interval $[-1, 1]$, for Rabi frequencies $\Omega_{P/S}$, we rescale this on the output range $[0, 1]$ such that all amplitudes $\Omega_{P/S}$ are always positive and real, since phase changes are already considered by the optimisation of $\Delta_{P/\delta}$. No analogous rescaling is performed for detunings $\Delta_i$. Thereafter, we rescale any action (in what follows any action, either amplitude $\Omega_i$ or detuning $\Delta_i$ is defined as $a_i$) by the maximum Rabi frequency $\Omega_{max}$ or maximum detuning $\Delta_{max}$.

We apply additional smoothing and rescaling operations to ensure the agent discovers experimentally realistic pulses. The time-scale of the dynamics simulation of fixed to some finite value, namely $1\mu s$ for the $\Lambda$ system, $0.5\mu s$ the Rydberg atom and $0.2\mu s$ for the transmon. In turn all control signals are defined in units of MHz, both $\Delta_{P/\delta}$ and $\Omega_{P/S}$ are divided into 50 timesteps for the $\Lambda$ system and Rydberg atom and 100 timesteps for the transmon. This gave a good tradeoff between signal expressiveness and speed.

The actions $a_i$ are smoothed with a Gaussian convolution $(a * G)(t) = \int_{-\infty}^{\infty} \mathcal{A}(\tau)G(t - \tau)\, d\tau$, where the Gaussian function $G(t)$ is defined as:

$$G(t) = N(t_\sigma) \exp\left(-\frac{t^2}{2 \cdot t_\sigma^2}\right), \tag{20}$$

where $t_\sigma$ defines the standard deviation and its value corresponds to the strength of the convolution filter. An ablation over this is provided in Fig. 2. This ensures that the generated time-dependent control signals are smooth and give rise to dynamics which can be solved in fixed number of timesteps, particularly at the beginning of the learning process when signals are randomly initialised. Pulse amplitude ends are always fixed at zero to ensure experimental viability with finite rise time effects, as signals cannot instantaneously start at non-zero amplitudes. Additionally, we use cubic spline interpolation (or linear interpolation for the transmon) between action samples which is efficient for use with adaptive step size solver used for solving the GKSL master equation in different environments.

The pulse smoothness is defined in terms of different pulse smoothness functions. The first smoothing function is constructed by calculating the second derivative of $\mathcal{A}(t)$:

$$S_{der}(a(t)) = \int_0^1 \left(\frac{d^2\mathcal{A}}{dt^2}\right)^2 dt. \tag{21}$$

An alternative smoothing function is defined in terms of the difference in output to that generated by a low pass Butterworth filter (Butterworth, 1930). This requires an expression of the filtered action which is the convolution of $\mathcal{A}(t)$ with the impulse response $h(t)$ of the Butterworth filter:

$$a_{\text{filtered}}(t) = (h * A)(t) = \int_0^1 h(t - \tau) A(\tau) \, d\tau \tag{22}$$

Calculating the difference with respect to the unfiltered signal, we get an expression for the low-pass smoothness with respect to a cutoff frequency $\omega_{max}$ and the filter order $n_{order}$:

$$S_{lp}(a(t), n_{order}, \omega_{max}) = \int_0^1 \left[ \int_0^1 h(t - \tau) a(\tau) \, d\tau - a(t) \right] dt. \tag{23}$$

It shall be noted that since all signals are discretised, the integrals decompose into discrete sums. The reference smoothness for an action is given by $S(B(t))$, where $B(t)$ is the Blackman window comprised of $n$ samples where $n$ also defines the number of signal samples corresponding to $\Omega_i$ / $\Delta_i$, which reads:

$$B[n] = \begin{cases} 0.42 - 0.5 \cos\left(\frac{2\pi n}{N-1}\right) + 0.08 \cos\left(\frac{4\pi n}{N-1}\right), & 0 \leq n \leq N - 1 \\ 0. & \text{otherwise} \end{cases} \tag{24}$$

This choice is made as it is designed to have minimal spectral leakage, which means it suppresses high-frequency components effectively and mimics the smoothness of the signals that we are looking for. Penalising pulse smoothness is required because even after applying a convolution filter, we do not attain signals which exhibit low enough smoothness. The importance of generating "smooth" functions is three-fold: firstly smoother waveforms are easier to experimentally implement with electronics with limited instantaneous bandwidth, as well as finite modulator rise times, and they are less vulnerable to signal chain delay or timing issues. Secondly, they are more interpretable in terms of the time evolution of the different quantum states. Thirdly, increased smoothness significantly speeds up the adaptive step size solver time which is particularly advantageous when working with limited computational resources or larger quantum systems.

Choosing the right smoothness penalty in the construction of the reward function is important as it can determine the learning speed and the extent to which realistic and interpretable controls are generated. We find, that a low-pass filter approach with the right cutoff frequency generally works well and provides the fastest learning of "smooth signals" as shown in Fig. 13. Other simpler smoothness functions such as the $L_1$ or $L_2$ norm are not considered because they were less well adapted for finding smooth signals that solved the quantum dynamics problems with a finite number of maximal adaptive solver steps.

Picking the right hyperparameters for the Gaussian convolution filter standard deviation $t_\sigma$ defined in equation 20, as well as the right smoothing penalties $w_\Delta$ and $w_\Omega$ (cf. equation 7) is crucial to ensure the optimal trade-off between smooth signal discovery to facilitate parallel optimisation, improved interpretability and discovery of high fidelity solutions. Overly strong signal smoothing or smoothing penalties result in the optimiser focussing largely on signal smoothness over fidelity of the quantum control task which is the primary objective. This is shown clearly in Fig. 2, where the $\Lambda$ system benefits from higher strict smoothing in form of a larger Gaussian kernel and higher weak smoothing in form of a larger pulse smoothness penalty, compared to the two photon Rydberg gate.

A final objective which competes with the fidelity, are the pulse areas $A(\Omega_i)$ and implicitly the pulse duration. $\Omega_{max}$ is limited physically by laser, RF or microwave power. Additionally, minimising pulse area is important for reducing the pulse energy and in turn the amount of heat introduced into the system, particularly for those quantum systems operating at cryogenic temperatures. Generally faster pulse sequences increase the clock cycles of a particular quantum operation which is desirable, but secondary to their fidelity, so implementing optimal control for some maximal amplitude $\Omega_{max}$ but with a minimal pulse area is considered in the example of a $\Lambda$ system. The baseline pulse area (cf. equation 7), which is particularly relevant for the results shown in Fig. 7 and Fig. 3 is computed by comparing the generated pulse area $A = \int_{t=0}^{t=1} \Omega(t) dt$ to the area of a Blackman window $A_B$ defined over the same timescale.

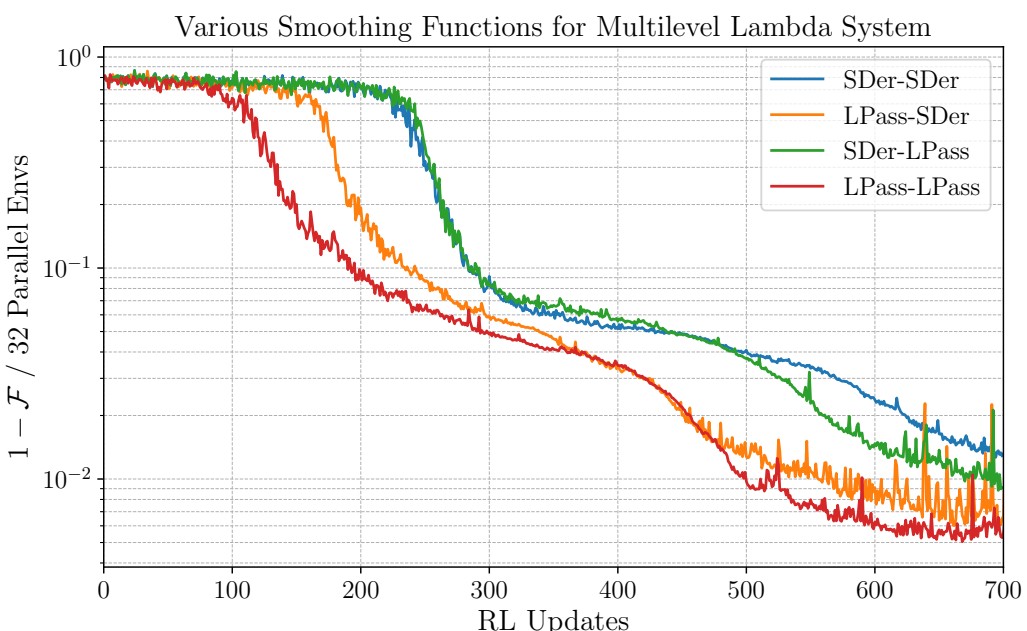

Figure 13: Comparison of different smoothing functions for mean infidelity (1- $\mathcal{F}$) across 32 parallel environments with different seeds plotted against the number of RL Updates for multi-level Lambda system. The legend corresponds to the type of smoothness penalty used where the ordering of the labels describes the amplitude and detuning smoothness functions respectively. One can observe that particularly for the amplitudes $\Omega_i$, using a low pass filter (LPass cf. equation 23) instead of a second derivative penalty (SDer cf. equation 21) allows for significantly sped up learning and also higher mean fidelity. For this ablation, the smoothness penalties $w_\Omega = w_\Delta$ are fixed to 0.001.

### E.3 NOISE MODEL

We use an Ornstein-Uhlenbeck noise model defined with standard deviation $\sigma$ and mean $\mu$ which defines time-dependent noise in time $t$:

$$\nu_t = \nu_{t-1}(1 - \alpha^2) + \sqrt{2}\sigma X(t)\alpha + \sigma^2\mu, \tag{25}$$

where $\alpha$ defines the characteristic time scale of the noise fluctuations and $X(t)$ is a random Gaussian noise at time $t$ with a standard deviation of 1 and a mean of 0.

