# OpenReview forum: "Scaleable Quantum Control via Physics Constrained Reinforcement Learning"
_ICLR.cc/2025/Conference — Submitted to ICLR 2025_

### Official Review · Reviewer_1yja · 2024-10-28

**Soundness:** 2
**Presentation:** 2
**Contribution:** 2
**Rating:** 5
**Confidence:** 3

**Summary:**

This paper provides an approach to RL for open quantum control, evaluated on 3 quantum systems with a focus on hardware realistic control. They demonstrate SotA fidelity using this approach.

**Strengths:**

- The promise of open source code is good and beneficial to reproducibility
- More realistic quantum control is an important research direction
- The superconducting results are especially compelling

**Weaknesses:**

- Minor mistakes, e.g. “quantu system”
- Figure 1 is the main selling point, but could be improved. There is an extra large space under the figure caption, I assume from latex formatting. The labels should be consistent, it’s called “infidelity” in the caption, but “fidelity” in the plot (obviously these are trivially related quantities, but just keeping the labeling consistent is worthwhile). I’m not sold on the one axis being log and the other y axis being non-log, it doesn’t make things super clear. Especially given that the log plot is basically just line going to small value (and it doesn’t bounce around a lot at these small values to make showing them super important). Also, Figure 1 would benefit from putting the legends in the same text block. All in all, I think there is a lot of room to make this a more compelling “Figure 1”
- The RL aspect of this paper is sometimes unclear. Reading from the beginning seems to indicate that bandit based algorithms would be the focus (but there isn’t much mention of bandit literature or the algorithms at play there). However, in 5.0.1, PPO is mentioned which doesn’t seem to come up again until 5.4. Clarifying this for the reader would be beneficial. Perhaps even explicitly writing out an algorithm block (could be in the appendix).
- Footnotes 2, 3 seem to be broken in Table 1
- Table 1 provides means, but no uncertainty/range on these estimates
- Figure 9 seems lower resolution/DPI than the other figures
- Two difficulties are mentioned in sec 4.1, which largely reduce down to 1) quantum simulation is hard and 2) RL is hard. While 2) is addressed through parallelism, 1) seems like a more important hurdle to this work and doesn’t seem to be addressed.
- Execution of the new superconducting pulses on real devices would be very interesting (but I recognize this may not be a realistic request)
- Ablation studies are lacking on the RL side (PPO vs other algorithms, parameters of PPO, etc.)

**Questions:**

- If the assumption is made of a white box simulator (here being an ODE), couldn’t one just directly optimize under constraints by differentiating through the solver? What’s the advantage of RL in cases of quantum control where we have perfect access to a differentiable simulator?

- To be clear, the RL agent (be it bandit or PPO) sees the entire quantum density matrix (“which is given as the density matrix”)?

---

> ### Author Response · Authors · 2024-11-22
> **Addressing R4 Weaknesses**
>
> We thank the reviewer for their detailed comments.
>
> We have addressed all minor mistakes.
>
> Regarding Fig. 1, we thank the reviewer for this feedback and have actioned it by colour coding the time and fidelity axes differently and removing the legend to avoid any confusion. Changing the x axis to log is not practical , given the ranges of max steps we ran. We have also added an explicit definition of infidelity in the figure caption and added uncertainty in GPU time to improve the overall presentation.
>
> We would like to point out that the standard bandit setting assumes a discrete action space, while our environments have a continuous action space. We further clarified this in the manuscript and also added an explicit definition of our bandit setting and PPO in Appendix Section A.
>
> Regarding footnotes - Thank you for this remark. We have addressed this issue.
>
> The comment about Tab. 1 is an important point we have addressed and we are grateful to the reviewer for pointing this out. Means/uncertainty are not relevant for the analytic solution and cannot be provided for the RL solution in Ref. Norambuena 2023, as no code was given for reproducing this experiment. We have however provided uncertainties for the other approaches and added an improved result for our own algorithm by incorporating ablations as suggested by Reviewer 1. We would like to emphasise that this also shows the enhanced robustness of our methodology to changes in the random seed and found similar observations when trying out direct differentiation of the action given to the solver for the transmon system.
>
> Thank you for the remark regarding the resolution of Fig. 9, we have addressed this.
>
> Making quantum simulation truly easy would require a quantum computer. However, similar to the comment made by Reviewer 3 we address this explicitly by speeding up the quantum simulation significantly with a max solver step constraint where it is explicit in Fig. 1 that the biggest bottleneck for our algorithm is quantum simulation time and fewer number of maximum steps reduce computational time. If however signals are generated which cannot be solved by the solver this constraint is not useful for enhancing the generation of high fidelity solutions. Therefore, we add explicit action smoothing which is described in Sec. 4.2 and signal smoothing penalties (cf. Eq. 6) with various smoothing functions defined in App. Sec. E.2. Action smoothing and signal smoothing penalties implicitly reduce the number of required steps to solve the quantum dynamics problem allowing us to run more RL updates in finite time and allowing us to scale our approach to more difficult problems in larger state spaces like the Rydberg gate.
>
> This is a very relevant request and is exactly what we have been working on. We have not been able to get definite data from a real physical device yet, but aim to do this as soon as possible and are aiming to submit a follow up paper. We will update the reviewers in this regard if anything arises before the end of the deadline. An additional sentence about this is added in the conclusion.
>
> As PPO is among the best-performing and simplest RL algorithms, and as our contribution does not lie in the RL algorithm itself, we do not think that ablations against other algorithms would yield strong insights. If the reviewer thinks differently here, we could add additional experiments. We would further like to point out that our approach runs multiple hyperparameter configurations in parallel, to avoid the problem of RL algorithm’s sensitivity to the choice of hyperparameters.

---

> > ### Author Response · Authors · 2024-11-22
> > **Addressing R4 Questions**
> >
> > We thank the reviewer for their questions.
> >
> > The reviewer is perhaps thinking of a method we have described in the related works section called Gradient Ascent Pulse Engineering (GRAPE). This might work well for smaller, toy model systems, but it doesn’t scale to larger and more realistic systems. This approach also can not deal well with noise and does not devise solutions which are robust with respect to parameter fluctuations. The superiority of using PPO is that it can handle noise and devise more robust solutions for larger systems which is not possible by directly differentiating through the solver. The noise robustness of the solutions is explicitly shown in Fig. 7. The numerical stability of our method is also shown in Tab.1, where compared to an optimal control approach (similar to GRAPE)  which directly differentiates through the solver we can robustly learn an optimal solution without any dependency on the input seed. The Optimal Control Method will yield occasional high fidelity solutions, but will not reliably produce a good solution and also does not scale well to a larger number of signal parameters unlike PPO. We made similar observations for the Transmon environment where direct differentiation does not reliably yield high fidelity solutions.
> >
> > No, the agent only sees the density matrix at the final timestep for a given set of parameters. Therefore, methods solely relying on differentiation of the intermediate density matrix are not practical and also cannot be extended to real world experiments where this intermediate observation is not possible. We have clarified this in the main text in Sec. 3.2. We also incorporate the average excited state population during the simulation into the state, for the Lambda and Rydberg system for improving learning dynamics. In Sec. 5.4 in the multi-step formulation, we only sample classical control noise which is used to adjust the policy, but this does not require any knowledge of the density matrix at intermediate steps.

---

> > > ### Comment · Reviewer_1yja · 2024-11-23
> > >
> > > I appreciate the clarification on the relationships between differentiating through solver methods and this approach, I believe the main text would also benefit from an expansion/clarification as outlined here.

---

> > > > ### Author Response · Authors · 2024-11-24
> > > > **Clarification on including comparison between our approach and direct differentiation in main text**
> > > >
> > > > Thank you for your thorough feedback and for highlighting the importance of clarifying the relationship between our approach and differentiating through solver methods. Based on your suggestion, we have added a dedicated **Table 1** explicitly comparing our RL approach to direct differentiation and explaining the relative strengths and weaknesses. Additionally, we have incorporated benchmarks for direct differentiation for every problem we considered. These can be found in **lines 357, 419, 463.**
> > > >
> > > > We are committed to addressing all concerns and making the manuscript as clear and comprehensive as possible. We would be grateful if you could provide **further clarification on the reasons your score remains below the acceptance threshold**.
> > > >
> > > > Best wishes,
> > > > the authors

---

> > ### Comment · Reviewer_1yja · 2024-11-23
> >
> > I appreciate the authors feedback. I believe the collection of improvements they have described will improve the paper, and I have adjusted my score accordingly.

---

### Official Review · Reviewer_LoSq · 2024-11-01

**Soundness:** 2
**Presentation:** 2
**Contribution:** 2
**Rating:** 3
**Confidence:** 3

**Summary:**

The paper presents a scalable quantum control framework utilizing physics-informed reinforcement learning (RL) to address the challenge of optimizing control signals for complex quantum systems. The authors devise an RL algorithm that incorporates physical constraints to restrict the solution space, focusing on desired time scales of quantum state dynamics and realistic control signal limitations.  The method is evaluated on three quantum systems—multi-level Λ systems, Rydberg atoms, and superconducting transmons—demonstrating higher fidelities and robustness to perturbations compared to previous methods.

**Strengths:**

- This manuscript introduces a novel framework that combines physics-informed constraints with reinforcement learning for quantum control. The use of physics-based constraints not only enhances solution quality but also ensures that the control signals are physically realistic and experimentally feasible.
- The paper provides a thorough evaluation of the method across different quantum systems and under various conditions, which strengthens the credibility of the results.

**Weaknesses:**

The main concern is that I do not consider ICLR the appropriate venue for this paper, as it does not provide the necessary background information for readers unfamiliar with quantum control. It would be more suitable for the paper to be published in a physics journal rather than an AI conference. Even though as a paper for physics journal, it still lacks the self-contained nature and clarity necessary for a comprehensive understanding by the readers. For instance, the experiments presented in this paper appear to be related  to a pre-existing methodology known as 'STIRAP'. However, the manuscript fails to provide adequate explanation or details regarding this method, which could leave readers without a clear understanding of its role and significance in the study.

Furthermore, I am not convinced that the results presented in this paper sufficiently substantiate the authors' claims. Specifically, the authors claim "the applied constrained not only improves computational efficiency but also promotes the selection of more physically realistic control signals." However, the manuscript does not offer adequate numerical or theoretical evidence to validate this assertion.

**Questions:**

Please see "Weaknesses" above.

---

> ### Author Response · Authors · 2024-11-22
> **Addressing R3 Weaknesses**
>
> We thank the reviewer for their comments.
>
> We have improved the explanation of subject specific terms, such as rise time and state population as also requested by Reviewer 1 and also explicitly define what it means for a process to be coherent or not in the background on quantum dynamics. STIRAP, which refers to Stimulated Raman Adiabatic Passage, is a widely studied quantum population transfer technique in physics, but we understand that many readers might not be familiar with it. Understanding the details of this methodology, is however not crucial for understanding the main results of the paper. Therefore, we have left the acronym out from the main text to make it more readable to a non-physics audience. A detailed explanation is provided in App. Sec. B for the interested readers. We hope this addresses the reviewers' concern, but are open for further feedback.
>
> We agree with the referee that proposals for the improvement of particular physical procedures would be better placed in physics journals. However, this paper is showing how to adapt established RL methods to better deal with real-world problems that are subject to noise and other constraints which are normally left aside. This is widely applicable to many problems, and by no means restricted to the few examples we have chosen for illustration. The paper presents how to adapt RL methods to better tackle real-world problems, and we are therefore convinced that the ICLR is the appropriate venue.
>
> We would like to point the reviewer to Figure 1, where we show that our developed approach yields improvements in both computational efficiency (i.e. lower compute time than in the case of higher max steps across all environments) and in solution quality i.e. higher fidelity than in higher max step case for the Rydberg environment, or no increases in solution quality as compute time increases for the Lambda system and the Transmon. We hope that this addresses the reviewer’s concern, and otherwise politely ask for further clarification.

---

> ### Author Response · Authors · 2024-11-23
> **ICLR appropriate venue for this paper**
>
> Dear Reviewer,
>
> Regarding your main concern, we wanted to highlight that the ICLR call for papers **explicitly calls for papers on "applications to physical sciences (physics, chemistry, biology, etc.)"** (see https://iclr.cc/Conferences/2025/CallForPapers). Further, papers of similar scope were presented at prior AI conferences (see, e.g., https://proceedings.neurips.cc/paper_files/paper/2022/file/1e70ac91ad26ba5b24cf11b12a1f90fe-Paper-Conference.pdf).
>
> We would be grateful if you could **assess our paper purely based on its quality and contributions** and update your score accordingly.
> If you remain doubtful whether ICLR is a suitable venue, please raise this in a follow up comment.
>
> Best wishes,
> The authors

---

> > ### Comment · Reviewer_LoSq · 2024-12-02
> >
> > I appreciate the authors' efforts, but I stand by my original assessment. While I acknowledge that papers on AI applications to physical sciences are an important subject at ICLR, I believe clear preliminary explanations are necessary. This paper is filled with physical terms that most readers at ICLR are unfamiliar with. For example, much of the experiments section is devoted to describing physical settings rather than discussing the RL methods and their functionality. Therefore, I do not consider it to be a well-presented AI paper and still believe that a physics journal would be a more appropriate venue for this work. In other words, in its current form, this paper obviously contributes more to the field of physics rather than to computer science.

---

### Official Review · Reviewer_Nyjc · 2024-11-02

**Soundness:** 3
**Presentation:** 4
**Contribution:** 2
**Rating:** 6
**Confidence:** 2

**Summary:**

This paper frames quantum control as a bandit problem and applies reinforcement learning to address it. It introduces a physical constraint as a reward term to penalize the required number of quantum simulation steps. The paper verifies the effectiveness of the approach on three problems, additionally showcasing the potential advantage of multi-step RL.

**Strengths:**

- The paper is well-written and easy-to-follow.
- It is sound to take advantage of RL's search capability to solve optimization problems.
- The implementation of GPU-accelerated parallel simulation is promising and useful.
- The experiment is comprehensive and informative.

**Weaknesses:**

- The contribution should be clarified and clear. It seems that the physical constraint is part of reward shaping, making the primary contribution of the paper the development of GPU-based simulation and the design of reward functions.
- The paper lacks theoretical results, though this is understandable given its focus on an RL application; therefore, this is only a minor weakness.

**Questions:**

- What is the MDP formulation in the multi-step RL case?
- Could the authors elaborate further on the comparison between multi-step RL and bandit approaches, and explain why multi-step RL provides a performance advantage?

---

> ### Author Response · Authors · 2024-11-22
> **Addressing R2 Weaknesses**
>
> We thank the reviewer for their feedback.
>
> We would like to draw the reviewers attention to our contributions listed at the end of the Introduction and ask for further clarification on what is unclear.
>
> Regarding the point on theoretical results, the examples discussed here have been subject to a large number of optimisation attempts using a variety of analytic and numerical methods throughout the last 10 -20 years. The efficiencies we seem to achieve with our new approach outperform these past results significantly. We consider this a theoretical breakthrough, for as long as the experimental verification, which we are working on, is outstanding.

---

> ### Author Response · Authors · 2024-11-22
> **Addressing R2 Questions**
>
> We thank the reviewer for their questions.
>
> We formulate the problem as a four-step markov decision process. After each step, the agent observes the mean signal noise and chooses the control signal for the next 0.25 microseconds. We have made this more explicit in App. Sec. A.2.
>
> We would like to point out that our setup differs from standard bandit settings, as it has a continuous action space. We have clarified this in the manuscript. The multi-step framework provides benefits if the agent is able to observe relevant additional information, that is not present at the start,  throughout the control process. In the studied the scenario, the agent observes the control signal noise throughout the process and is able to adjust its action accordingly. Note that this is possible as control signal noise often has a large bias, which the agent can adapt its control signal to.

---

> > ### Author Response · Authors · 2024-12-02
> > **Follow up to with reviewer**
> >
> > Dear Reviewer,
> >
> > We thank you for reviewing our manuscript. We were wondering whether you had the chance to look at our comments and the revised manuscript - we incorporated several suggestions, allowing us to improve our results and the presentation of our work.
> >
> > We would be grateful if you could suggest any further improvements or adjust your score.
> >
> > Best regards,
> > The Authors

---

### Official Review · Reviewer_8oHM · 2024-11-03

**Soundness:** 3
**Presentation:** 3
**Contribution:** 2
**Rating:** 6
**Confidence:** 4

**Summary:**

Contributes in the field of optimal quantum state control, where the implementation of optimal external signals to quantum systems is difficult for real and noisy systems. The key idea is the use of physical information about the quantum state dynamics as priors to help the RL agent as constrains. The algorithm then learns a control policy that maximizes the fidelity of the quantum control task by reducing/removing control signals which result in overly fast quantum state dynamics, which helps with maintaining a better controllable system state. The approach is tested on three well known problems in the control literature and shown empirically to achieve higher fidelities than a set of baseline algorithms and robustness w.r.t. to state perturbations.

**Strengths:**

Three well motivated application examples. Short but good baseline study in table 1, with good statistical averages of 32 random seeds. Very clearly written, well referenced, brief but concise descriptions which were mostly understandable even from an outside perspective. Results appear sound and significant from what I understand. I think this is a generally solid paper with a sound contribution.

**Weaknesses:**

- A lot of use of quite specific jargon, which makes it harder to follow and otherwise very clearly written paper.

- The section on multi-step RL is very short and by only referencing results in the appendix, not self-contained material of this paper. The introduction (L 84) claims an “investigation to address larger levels of system noise” which is only pointed to / contained in the appendix.

- RL ‘framework’ is a very broad claim, a physical reward shaping with (as I understand) out-of-bound-step constraints may be effective as application to quantum state control, but is - in the context of RL and QRL - not very novel.
---
Minor Notes:

- Fig 1. Axis labels should be bigger. Split positioning of legend is a bit confusing.

- The terms state populations, rise time etc. could be better explained.

- L163 Typo Two dots after etc.

- L 202 Typo quantu[m] system, see [Section] 3.2.1)

- L 350 Typo That the learned pulses [are] physically

- L 350 Typo e.g.[,]

- Inconsistent use of references (eq / equations), (Fig. / Figure), (App. / Appendix)

- Inconsistent use of cf. inline and (cf. in brackets)

**Questions:**

- Is there a reason that the multi-step return does not include a discount factor as is usually the case?
- L. 260 What is a baseline ‘blackman envelope’?
- What differentiates the physically feasible control signals in Tab. 1?
- RL in general is known to be quite sensitive against reward shaping attempts (resulting in reward hacking instead), and some of the reward terms in Eq 6 seem just added for good measure. Was an ablation study done on these terms and components (ReLU, blackman envelope, the smoothness estimator etc.)?
- Although PPO is a solid choice in many RL applications, since the action-space is very simple, small and discrete, DQN or even simpler policy gradient methods may already be sufficient. Have you perhaps tested any ablation / combination of other RL approaches and could elaborate on their comparative performance?

---
All questions have been insightfully addressed by the authors in the rebuttal. As a result the confidence score has been updated from 3 -> 4.

---

> ### Author Response · Authors · 2024-11-22
> **Addressing R1 Weaknesses**
>
> We thank the reviewer for their constructive feedback.
>
> Regarding the use of subject specific jargon we have tried to remove jargon or explicitly define it wherever possible. Rise time, coherent and state population are now explicitly defined. Any reference to acronyms like STIRAP are removed from the main text and detailed explanations are relegated to the Appendix to make the paper more readable and to signpost readers who are interested in more detail.
>
> We have added a sentence to explain the results in the section on multi-step RL with a reference to the Appendix for further details, but note that we considered the results in the previous sections more significant, especially in relation to them outperforming prior work significantly and prioritised placing these figures in the main body of the text.
>
> The wording of our method as an 'RL Framework' is a valid concern, we reworded this from framework to implementation.
>
> We thank the reviewer for their attention to detail and have incorporated all these changes, including an explicit definition of rise time (Sec. 2) and state population (Sec. 3.1) . Fig. 1 was also modified to improve readability across the split legend with improved colour coding, standard deviation in GPU times across RL updates and bigger axes labels.

---

> > ### Author Response · Authors · 2024-11-22
> > **Addressing R1 Questions**
> >
> > We are extremely grateful for the reviewers questions as reviewing them allowed us to improve our results and achieve over 0.999 fidelity across all considered examples.
> >
> > Regarding the discount factor, as the environment has a fixed number of steps, we did not include a discount factor. A discount factor is only required if the number of steps is unbounded, or if there is a reason to prefer immediate rewards, which is not the case here. We have made this more explicit in App. Sec. A.2.
> >
> > A blackman envelope is defined in App. Eq 19., but it is not crucial for the explanation of reward shaping and therefore no explicit reference is made to it in the main text anymore, further details on this are provide in App. Sec. E.2.
> >
> > Some signals do not start and end at zero amplitude, they might be “theoretically ideal” but this is unphysical and we have made this explicit in the text. Furthermore, discontinuities or changes in control parameters exceeding the bandwidth of a real system cannot be achieved and must be discarded.
> >
> > The question on the reward function is a very important observation and we are very grateful to the reviewer for their comment. There was already an ablation over different smoothing estimators contained in App. Fig. 13  which showed that the lowpass filter smoothness calculation led to the most optimal learning dynamics. We performed additional ablations shown  in App. Fig. 12 and actually found better solutions for both the Lambda system problem, as well as the Rydberg system than before now exceeding 0.999 fidelity. An ablation over the area penalty is found for the signals in Fig. 2, as well as for the App. Figs. 5 and 6. We have also simplified the description of the reward function and relegated one system specific term to the relevant Lambda System and Rydberg sections of the Appendix to improve general readability. We note that the smoothing terms, in combination with the formulation as a constrained RL problem, play an important role in limiting the time dynamics of our solution and allow us to outperform prior methods significantly, as picking the wrong smoothing related parameters has a significant effect on solution quality as shown in App. Figs. 12 and 13.
> >
> > We would like to point out that the action space is continuous and not discrete (which has been clarified in the main text), hence DQN may not be a favourable choice. As PPO is among the best-performing and simplest RL algorithms, and as our contribution does not lie in the RL algorithm itself, we do not think that ablations against other algorithms would yield strong insights. If the reviewer thinks differently here, we could add additional experiments.

---

> > > ### Comment · Reviewer_8oHM · 2024-11-26
> > >
> > > First of all I’d like to thank you for these insightful replies, and all your efforts in providing extra results. Some thoughts to your explanations:
> > >
> > > - I agree with your assessment regarding the discount factor, four steps do not warrant a time/step discount. From the initial draft I was assuming “multi-step” to be more in the range of 10+ steps, from which onwards the optimization of fewer control actions (as reason to prefer immediate rewards) would have perhaps benefitted from an experimental check.
> > > - Also thank you for the clarification on the continuous action space, similarly to reviewer 1yja, I associate bandit problems usually with discrete choices (regarding the levers=actions to pull). In hindsight it does make sense for the control pulse actions to be of continuous nature, but the updated draft now makes this nice and clear. From this assumption also stems the question regarding the DQN, but with continuous actions I also think PPO is a valid initial choice for RL practitioners after all.
> > > - I’m happy to hear that the question prompted another improvement, I think the ablation also turned out quite insightful. I find Figure 12 on the ablation of pulse smoothing penalties in particular to be a quite nice result, perhaps if space/formatting permits this would make a nice plot in the main paper for reasoning about the reward shaping components?
> > > - The addition of variances in the Figure 1 and Table 1 are a great details to include. I also skimmed over the updated draft and I indeed find the jargon to be more approachable now.
> > > - I’d also agree with reviewer 1yja that open-sourcing this project would be a very welcome contribution to the quantum simulation community.
> > >
> > > Ultimately I think the paper got better with this rebuttal and deserves the positive leaning score. I have added a note in the review that the questions have been addressed and updated my confidence score +1.

---

> > > > ### Author Response · Authors · 2024-11-28
> > > > **Replying to comment & final paper revision**
> > > >
> > > > Thank you for your helpful and detailed feedback and for recognising the improvements made to our paper following the rebuttal. We have added the suggested ablation study to the main text as Fig. 2 which now includes all environments and have included an explanation in Section 4.2.
> > > >
> > > > Since you mentioned that the paper has improved and merits a positive score, we were wondering whether there are any remaining concerns that result in the "just positive" rating? If there are any additional areas that need clarification or improvement, we are happy to address or discuss them.
> > > >
> > > > Best regards, the authors

---

> > > > > ### Author Response · Authors · 2024-11-28
> > > > > **Open Sourcing Code**
> > > > >
> > > > > We completely agree that open sourcing the code would be a great contribution to the quantum simulation community. We will publicise our code with detailed documentation on GitHub once the anonymous review process is completed.
> > > > >
> > > > > If the reviewer thinks this would be helpful, we can also share code examples anonymously here.

---

### Meta-Review · Area_Chair_GpFy · 2024-12-12

**Metareview:**

The paper considers a class of optimal quantum control problem, and the authors propose to solve this problem via reinforcement learning (RL) method. A reward shaping approach is proposed to handle the physical constraints of the problem. For experiments, the paper considers three quantum control problems that can be viewed as RL under the bandit setting or an episodic setting (with horizon 4).

The main issue of this paper is 2-fold.

First, several reviewers have pointed out that the paper is full of jargons and is hard to read. As the AC is not in the area of quantum computing, the AC is very careful on this complaint. We carefully check the background of the reviewers and find all of them having relevant publications in the quantum area. And we confirm that this complaint is not because the reviewers are in irrelevant areas. The AC has also searched several past ICLR publications about quantum topics. We do find the past accepted papers much easier to read and they all incorporate limited physics jargons.

Second, in terms of reinforcement learning, the contribution of the paper is limited. This is because, if we remove the physics background, a bandit or an episodic MDP with horizon 4 is not a challenging task in RL. Moreover, the constraint is simply handled by reward shaping, which is also quite common. The paper primarily adapts existing RL techniques to specific quantum control problems and hence lacks enough novelty.

Therefore, the AC decides to reject this paper.

**Additional Comments On Reviewer Discussion:**

Most of the concerns from the reviewers are relatively minor. But there are two key issues that are not well-addressed.

First, the paper has too much jargons. The is pointed out by several reviewers. Despite the effort of the authors, we do not think their changes successfully resolve this issue. The paper is still full of physics jargons that are hard to understand by the ICLR audiences.

Second, the contribution of in terms of RL methodology is not good enough (see meta-review). This is also pointed out by several reviewers and has not been fixed by the authors during the rebuttal.

---

### Decision · Program_Chairs · 2025-01-22

Reject